# DE NOVO MOLECULAR GENERATION VIA CONNECTION-AWARE MOTIF MINING

**Zijie Geng**[1]*, **Shufang Xie**[2]†, **Yingce Xia**[3]†, **Lijun Wu**[3], **Tao Qin**[3], **Jie Wang**[1,4]†,
**Yongdong Zhang**[1], **Feng Wu**[1], **Tie-Yan Liu**[3]

[1] University of Science and Technology of China
`ustcgzj@mail.ustc.edu.cn`, `{jiewangx, zhyd73, fengwu}@ustc.edu.cn`
[2] Gaoling School of Artificial Intelligence, Renmin University of China
`shufangxie@ruc.edu.cn`
[3] Microsoft Research AI4Science
`{yingce.xia, lijunwu, taoqin, tyliu}@microsoft.com`
[4] Institute of Artificial Intelligence, Hefei Comprehensive National Science Center

## ABSTRACT

*De novo* molecular generation is an essential task for science discovery. Recently, fragment-based deep generative models have attracted much research attention due to their flexibility in generating novel molecules based on existing molecule fragments. However, the *motif vocabulary*, i.e., the collection of frequent fragments, is usually built upon heuristic rules, which brings difficulties to capturing common substructures from large amounts of molecules. In this work, we propose a new method, MiCaM, to generate molecules based on mined connection-aware motifs. Specifically, it leverages a data-driven algorithm to automatically discover motifs from a molecule library by iteratively merging subgraphs based on their frequency. The obtained motif vocabulary consists of not only molecular motifs (i.e., the frequent fragments), but also their connection information, indicating how the motifs are connected with each other. Based on the mined connection-aware motifs, MiCaM builds a connection-aware generator, which simultaneously picks up motifs and determines how they are connected. We test our method on distribution-learning benchmarks (i.e., generating novel molecules to resemble the distribution of a given training set) and goal-directed benchmarks (i.e., generating molecules with target properties), and achieve significant improvements over previous fragment-based baselines. Furthermore, we demonstrate that our method can effectively mine domain-specific motifs for different tasks.

## 1 INTRODUCTION

Drug discovery, from designing hit compounds to developing an approved product, often takes more than ten years and billions of dollars (Hughes et al., 2011). *De novo* molecular generation is a fundamental task in drug discovery, as it provides novel drug candidates and determines the underlying quality of final products. Recently, with the development of artificial intelligence, deep neural networks, especially graph neural networks (GNNs), have been widely used to accelerate novel molecular generation (Stokes et al., 2020; Bilodeau et al., 2022). Specifically, we can employ a GNN to generate a molecule iteratively: in each step, given an unfinished molecule $\mathcal{G}_0$, we first determine a new generation unit $\mathcal{G}_1$ to be added; next, determine the connecting sites on $\mathcal{G}_0$ and $\mathcal{G}_1$; and finally determine the attachments between the connecting sites, e.g., creating new bonds (Liu et al., 2018) or merging shared atoms (Jin et al., 2018). In different methods, the generation units could be either atoms (Li et al., 2018; Mercado et al., 2021) or frequent fragments (referred to as *motifs*) (Jin et al., 2020a; Kong et al., 2021; Maziarz et al., 2021).

For fragment-based models, building an effective motif vocabulary is a key factor to the success of molecular generation (Maziarz et al., 2021). Previous works usually rely on heuristic rules or tem-

---

*This work was done when Zijie Geng was an intern at Microsoft Research AI4Science.
†Corresponding author.

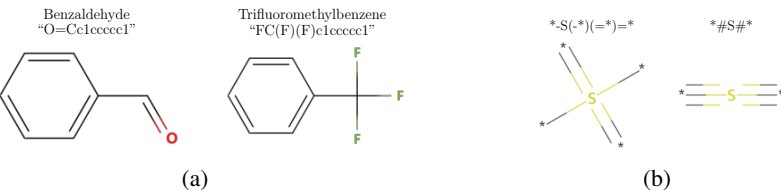

Figure 1: (a) Two substructures that occur frequently in ChEMBL. (b) Different connection modes of sulfur atoms. The "*-S(-*)(=*)=*" is commonly seen while"*#S#*" does not appear in ChEMBL.

plates to obtain a motif vocabulary. For example, JT-VAE (Jin et al., 2018) decomposes molecules into pre-defined structures like rings, chemical bonds, and individual atoms, while MoLeR (Maziarz et al., 2021) separates molecules into ring systems and acyclic linkers or functional groups. Several other works build motif vocabularies in a similar manner (Jin et al., 2020a; Yang et al., 2021). However, heuristic rules cannot cover some chemical structures that commonly occur yet are a bit more complex than pre-defined structures. For example, the subgraph patterns *benzaldehyde* ("O=Cc1ccccc1") and *trifluoromethylbenzene* ("FC(F)(F)c1ccccc1") (as shown in Figure 1(a)) occur 398,760 times and 57,545 times respectively in the 1.8 million molecules in ChEMBL (Mendez et al., 2019), and both of them are industrially useful. Despite their high frequency in molecules, the aforementioned methods cannot cover such common motifs. Moreover, in different concrete generation tasks, different motifs with some domain-specific structures or patterns are favorable, which can hardly be enumerated by existing rules. Another important factor that affects the generation quality is the connection information of motifs. This is because although many connections are valid under a valence check, the "reasonable" connections are predetermined and reflected by the data distribution, which contribute to the chemical properties of molecules (Yang et al., 2021). For example, a *sulfur* atom with two triple-bonds, i.e., "*#S#*" (see Figure 1(b)), is valid under a valence check but is "unreasonable" from a chemical point of view and does not occur in ChEMBL.

In this work, we propose **MiCaM**, a generative model based on **Mi**ned **C**onnection-**a**ware **M**otifs. It includes a data-driven algorithm to mine a connection-aware motif vocabulary from a molecule library, as well as a connection-aware generator for *de novo* molecular generation. The algorithm mines the most common substructures based on their frequency of appearance in the molecule library. Briefly, across all molecules in the library, we find the most frequent fragment pairs that are adjacent in graphs, and merge them into an entire fragment. We repeat this process for a pre-defined number of steps and collect the fragments to build a motif vocabulary. We preserve the connection information of the obtained motifs, and thus we call them *connection-aware motifs*.

Based on the mined vocabulary, we design the generator to simultaneously pick up motifs to be added and determine the connection mode of the motifs. In each generation step, we focus on a non-terminal connection site in the current generated molecule, and use it to query another connection either (1) from the motif vocabulary, which implies connecting a new motif, or (2) from the current molecule, which implies cyclizing the current molecule.

We evaluate MiCaM on distribution learning benchmarks from GuacaMol (Brown et al., 2019), which aim to resemble the distributions of given molecular sets. We conduct experiments on three different datasets and MiCaM achieves the best overall performance compared with several strong baselines. After that, we also work on goal directed benchmarks, which aim to generate molecules with specific target properties. We combine MiCaM with iterative target augmentation (Yang et al., 2020) by jointly adapting the motif vocabulary and network parameters. In this way, we achieve state-of-the-art results on four different types of goal-directed tasks, and find motif patterns that are relevant to the target properties.

## 2 OUR APPROACH

### 2.1 CONNECTION-AWARE MOLECULAR MOTIF MINING

Our motif mining algorithm aims to find the common molecule motifs from a given training data set $\mathcal{D}$, and build a connection-aware motif vocabulary for the follow-up molecular generation. It

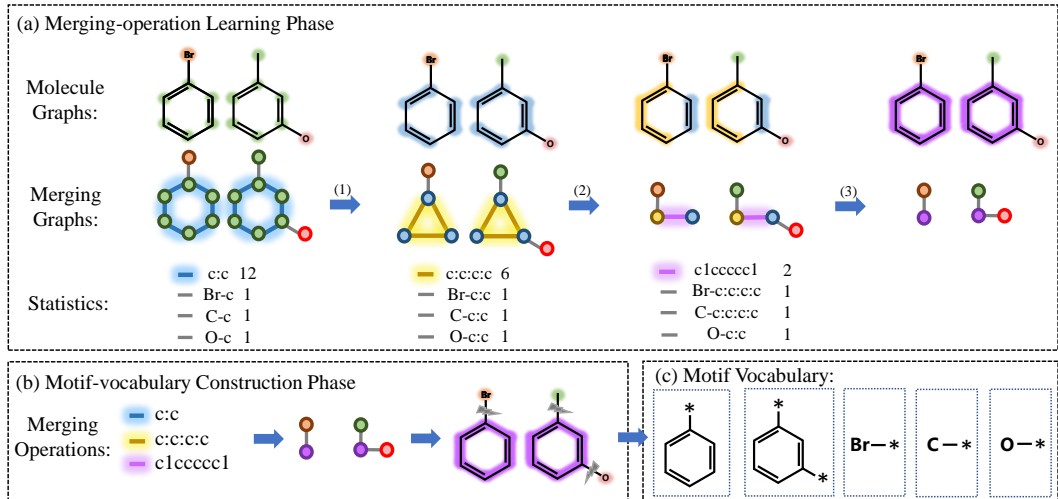

Figure 2: An example of connection-aware molecular motif mining. Given a training set $\mathcal{D} =$ {"Brc1ccccc1", "Cc1cccc(O)c1"}, it consists of two phases. (a) Merging-operation Learning Phase. The merging graphs are initialized the same as the molecular graphs. In the first iteration, "c:c" (marked in blue) is the most frequent pattern. We merge the patterns "c:c" in all the molecules to update the merging graphs. In the second iteration, "c:c:c:c" (marked in yellow) is the most frequent pattern, so we merge such patterns and update merging graphs. We repeat this process for 3 iterations and record the merging operations in order. (b) Motif-vocabulary Construction Phase. We apply the recorded merging operations sequentially on all molecules. Then the two molecules are fragmentized as motifs. We break the bonds between different motifs while preserving the broken bonds. In this way we construct a connection-aware motif vocabulary as shown in (c).

processes $\mathcal{D}$ with two phases: the merging-operation learning phase and the motif-vocabulary construction phase. Figure 2 presents an example and more implementation details are in Algorithm 1 in Appendix A.1.

**Merging-operation Learning Phase** In this phase, we aim to learn the top $K$ most common patterns (which correspond to rules indicating how to merge subgraphs) from the training data $\mathcal{D}$, where $K$ is a hyperparameter. Each molecule in $\mathcal{D}$ is represented as a graph $\mathcal{G}(\mathcal{V}, \mathcal{E})$ (the first row in Figure 2(a)), where the nodes $\mathcal{V}$ and edges $\mathcal{E}$ denote atoms and bonds respectively. For each $\mathcal{G}(\mathcal{V}, \mathcal{E}) \in \mathcal{D}$, we use a merging graph $\mathcal{G}_M(\mathcal{V}_M, \mathcal{E}_M)$ (the second row in Figure 2(a)) to track the merging status, i.e., to represent the fragments and their connections. In $\mathcal{G}_M(\mathcal{V}_M, \mathcal{E}_M)$, each node $\mathcal{F} \in \mathcal{V}_M$ represents a fragment (either an atom or a subgraph) of the molecule, and the edges in $\mathcal{E}_M$ indicate whether two fragments are connected with each other. We initialize each merging graph from the molecule graph by treating each atom as a single fragment and inheriting the bond connections from $\mathcal{G}$, i.e., $\mathcal{G}_M^{(0)}(\mathcal{V}_M^{(0)}, \mathcal{E}_M^{(0)}) = \mathcal{G}(\mathcal{V}, \mathcal{E})$.

We define an operation "$\oplus$" to create a new fragment $\mathcal{F}_{ij} = \mathcal{F}_i \oplus \mathcal{F}_j$ by merging two fragments $\mathcal{F}_i$ and $\mathcal{F}_j$ together. The newly obtained $\mathcal{F}_{ij}$ contains all nodes and edges from $\mathcal{F}_i$, $\mathcal{F}_j$, as well as all edges between them. We iteratively update the merging graphs to learn merging operations. In the merging graph $\mathcal{G}_M^{(k)}(\mathcal{V}_M^{(k)}, \mathcal{E}_M^{(k)})$ at the $k^{\text{th}}$ iteration ($k = 0, \cdots, K-1$), each edge represents a pair of fragments, $(\mathcal{F}_i, \mathcal{F}_j)$, that are adjacent in the molecule. It also gives out a new fragment $\mathcal{F}_{ij} = \mathcal{F}_i \oplus \mathcal{F}_j$. We traverse all edges $(\mathcal{F}_i, \mathcal{F}_j) \in \mathcal{E}_M^{(k)}$ in all merging graphs $\mathcal{G}_M^{(k)}$ to count the frequency of $\mathcal{F}_{ij} = \mathcal{F}_i \oplus \mathcal{F}_j$, and denote the most frequent $\mathcal{F}_{ij}$ as $\mathcal{M}^{(k)}$. [1] Consequently, the $k$-th *merging operation* is defined as: if $\mathcal{F}_i \oplus \mathcal{F}_j == \mathcal{M}^{(k)}$, then merge $\mathcal{F}_i$ and $\mathcal{F}_j$ together.[2] We apply the merging operation on all merging graphs to update them into $\mathcal{G}_M^{(k+1)}(\mathcal{V}_M^{(k+1)}, \mathcal{E}_M^{(k+1)})$. We repeat such a process for $K$ iterations to obtain a merging operation sequence $\{\mathcal{M}^{(k)}\}_{k=0}^{K-1}$.

---

[1] Different $(\mathcal{F}_i, \mathcal{F}_j)$ can make same $\mathcal{F}_{ij}$. For example, both (CCC, C) and (CC, CC) can make "CCCC".

[2] When applying merging operations, we traverse edges in orders given by RDKit (Landrum et al., 2006).

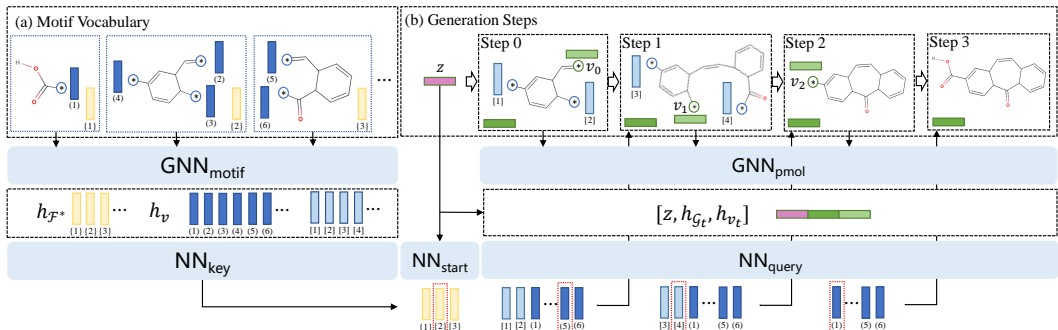

Figure 3: The Generation Steps. (a) The generation procedure is based on the connection-aware motif vocabulary. We obtain the graph representations $h_{\mathcal{F}^*}$ (yellow) and the node representations $h_v$ (dark blue) via $\text{GNN}_{\text{motif}}$. (b) In the $t^{\text{th}}$ generation step, we obtain the graph representation $h_{\mathcal{G}_t}$ (dark green) and node representations $h_v$ (light blue) via $\text{GNN}_{\text{pmol}}$. We focus on a connection site $v_t$ and use $[z, h_{\mathcal{G}_t}, h_{v_t}]$ to query another connection site either from the motif vocabulary (which implies adding a motif) or from the partial molecule (which implies cyclizing). The right answers of every steps are marked by red boxes.

**Motif-vocabulary Construction Phase** For each molecule $\mathcal{G}(\mathcal{V}, \mathcal{E}) \in \mathcal{D}$, we apply the merging operations sequentially to obtain the ultimate merging graph $\mathcal{G}_M(\mathcal{V}_M, \mathcal{E}_M) = \mathcal{G}_M^{(K)}(\mathcal{V}_M^{(K)}, \mathcal{E}_M^{(K)})$. We then disconnect all edges between different fragments and add the symbols "$*$" to the disconnected positions (see Figure 2(b, c)). The fragments with "$*$" symbols are connection-aware, and we denote the connection-aware version of a fragment $\mathcal{F}$ as $\mathcal{F}^*$. The motif vocabulary is the collection of all such connection-aware motifs: $\text{Vocab} = \cup_{\mathcal{G}_M(\mathcal{V}_M, \mathcal{E}_M) \in \mathcal{D}} \{\mathcal{F}^* : \mathcal{F} \in \mathcal{V}_M\}$. During generating, our model connects the motifs together by directly merging the connection sites (i.e., the "$*$"s) to generate new molecules.

**Time Complexity** The time complexity of learning merging operations is $O(K|\mathcal{D}|e)$, where $K$ is the number of iterations, $|\mathcal{D}|$ is the molecule library size, and $e = \max_{\mathcal{G}(\mathcal{V}, \mathcal{E}) \in \mathcal{D}} |\mathcal{E}|$. In practice, the time cost decreases rapidly as the iteration step $k$ increases. It takes less than 10 minutes and about 90 minutes to run $3,000$ iterations on the QM9 ($\sim 133K$ molecules) (Ruddigkeit et al., 2012) and ZINC ($\sim 219K$ molecules) (Irwin et al., 2012) datasets, respectively, using 6 CPU cores (see Appendix A.1). The time complexity of fragmentizing an arbitrary molecule $\mathcal{G}(\mathcal{V}, \mathcal{E})$ into motifs is $O(K|\mathcal{E}|)$, linear with the number of bonds.

## 2.2 MOLECULAR GENERATION WITH CONNECTION-AWARE MOTIFS

The generation procedure of MiCaM is shown in Figure 3 and Algorithm 2 in Appendix A.2. MiCaM generates molecules by gradually adding new motifs to a current partial molecule (denoted as $\mathcal{G}_t$, where $t$ is the generation step), or merging two connection sites in $\mathcal{G}_t$ to form a new ring.

For ease of reference, we define the following notations. We denote the node representation of any atom $v$ (including the connection sites) as $h_v$, and denote the graph representation of any graph $\mathcal{G}$ (either a molecule or a motif) as $h_{\mathcal{G}}$. Let $\mathcal{C}_{\mathcal{G}}$ denote the connection sites from a graph $\mathcal{G}$ (either a partial molecule or a motif), and let $\mathcal{C}_{\text{Vocab}} = \cup_{\mathcal{F}^* \in \text{Vocab}} \mathcal{C}_{\mathcal{F}^*}$ be the set of all connection sites from the motif vocabulary.

**Generation Steps** In the $t^{\text{th}}$ generation step, MiCaM modifies the partial molecule $\mathcal{G}_t$ as follows:

(1) *Focus on a connection site $v_t$ from $\mathcal{G}_t$.* We use a queue $\mathcal{Q}$ to manage the the orders of connection sites in $\mathcal{C}_{\mathcal{G}_t}$. At the $t^{\text{th}}$ step, we pop the head of $\mathcal{Q}$ to get a connection site $v_t$. The $\mathcal{Q}$ is maintained as follows: after selecting a new motif $\mathcal{F}^*$ to connect with $\mathcal{G}_t$, we use the `RDKit` library to give a canonical order of the atoms in $\mathcal{F}^*$, and then put the connection sites into $\mathcal{Q}$ following this order. After merging two connection sites together, we just remove them from $\mathcal{Q}$.

(2) *Encode $v_t$ and candidate connections.* We employ a graph neural network $\text{GNN}_{\text{pmol}}$ to encode the partial molecule $\mathcal{G}_t$ and obtain the representations of all atoms (including the connection sites) and the graph. The node representations $\boldsymbol{h}_{v_t}$ of $v_t$ and the graph representation $\boldsymbol{h}_{\mathcal{G}_t}$ of $\mathcal{G}_t$ will be jointly used to query another connection site. For the motifs $\mathcal{F}^* \in$ Vocab, we use another GNN, denoted as $\text{GNN}_{\text{motif}}$, to encode their atoms and connection sites. In this way we obtain the node representations $\boldsymbol{h}_v$ of all connection sites $v \in \mathcal{C}_{\text{Vocab}}$ [3]. The candidate connections are either from $\mathcal{C}_{\text{Vocab}}$ or from $\mathcal{C}_{\mathcal{G}_t} \setminus \{v_t\}$.

(3) *Query another connection site.* We employ two neural networks, $\text{NN}_{\text{query}}$ to make a query vector, and $\text{NN}_{\text{key}}$ to make key vectors, respectively. Specifically, the probability $P_v$ of picking every connection sites is calculated by:

$$P_v = \underset{v \in \mathcal{C}_{\text{Vocab}} \cup \mathcal{C}_{\mathcal{G}_t} \setminus \{v_t\}}{\text{softmax}} \left( \text{NN}_{\text{query}} \left( [\boldsymbol{z}, \boldsymbol{h}_{\mathcal{G}_t}, \boldsymbol{h}_{v_t}] \right) \cdot \text{NN}_{\text{key}}(\boldsymbol{h}_v) \right), \tag{1}$$

where $\boldsymbol{z}$ is a latent vector as used in variational auto-encoder (VAE) (Kingma & Welling, 2013). Using different $\boldsymbol{z}$ results in diverse molecules. During training, $\boldsymbol{z}$ is sampled from a posterior distribution given by an encoder, while during inference, $\boldsymbol{z}$ is sampled from a prior distribution.

For inference, we make a constraint on the bond type of picked $v$ by only considering the connection sites whose adjacent edges have the same bond type as $v_t$. This practice guarantees the validity of the generated molecules. We also implement two generation modes, i.e., greedy mode that picks the connection as $\arg \max P_v$, and distributional mode that samples the connection from $P_v$.

(4) *Connect a new motif or cyclize.* After the model picks a connection $v$, it turns into a connecting phase or a cyclizing phase, depending on whether $v \in \mathcal{C}_{\text{Vocab}}$ or $v \in \mathcal{C}_{\mathcal{G}_t}$. If $v \in \mathcal{C}_{\text{Vocab}}$, and suppose that $v \in \mathcal{C}_{\mathcal{F}^*}$, then we connect $\mathcal{F}^*$ with $\mathcal{G}_t$ by directly merging $v_t$ and $v$. Otherwise, when $v \in \mathcal{C}_{\mathcal{G}_t}$, we merge $v_t$ and $v$ together to form a new ring, and thus the molecule cyclizes itself. Notice that, allowing the picked connection site to come from the partial molecule is important, because with this mechanism MiCaM theoretically can generate novel rings that are not in the motif vocabulary.

We repeat these steps until $\mathcal{Q}$ is empty and thus there is no non-terminal connection site in the partial molecule, which indicates that we have generated an entire molecule.

**Starting** As in the beginning (i.e., the $0^{\text{th}}$ step), the partial graph is empty, we implement this step exceptionally. Specifically, we use another neural network $\text{NN}_{\text{start}}$ to pick up the first motif from the vocabulary as $\mathcal{G}_0$. The probability $P_{\mathcal{F}^*}$ of picking every motifs is calculated by:

$$P_{\mathcal{F}^*} = \underset{\mathcal{F}^* \in \text{Vocab}}{\text{softmax}} \left( \text{NN}_{\text{start}}(\boldsymbol{z}) \cdot \text{NN}_{\text{key}}(\boldsymbol{h}_{\mathcal{F}^*}) \right), \quad \boldsymbol{h}_{\mathcal{F}^*} = \text{GNN}_{\text{motif}}(\mathcal{F}^*). \tag{2}$$

## 2.3 Training MiCaM

We train our model in a VAE (Kingma & Welling, 2013) paradigm.[4] A standard VAE has an encoder and a decoder. The encoder maps the input molecule $\mathcal{G}$ to its representation $\boldsymbol{h}_{\mathcal{G}}$, and then builds a posterior distribution of the latent vector $\boldsymbol{z}$ based on $\boldsymbol{h}_{\mathcal{G}}$. The decoder takes $\boldsymbol{z}$ as input and tries to reconstruct the $\mathcal{G}$. VAE usually has a reconstruction loss term (between the original input $\mathcal{G}$ and the reconstructed $\hat{\mathcal{G}}$) and a regularization term (to control the posterior distribution of $\boldsymbol{z}$).

In our work, we use a GNN model $\text{GNN}_{\text{mol}}$ as the encoder to encode a molecule $\mathcal{G}$ and obtain its representation $\boldsymbol{h}_{\mathcal{G}} = \text{GNN}_{\text{mol}}(\mathcal{G})$. The latent vecotr $\boldsymbol{z}$ is then sampled from a posterior distribution $q(\cdot | \mathcal{G}) = \mathcal{N}\left( \boldsymbol{\mu}(\boldsymbol{h}_{\mathcal{G}}), \exp(\boldsymbol{\Sigma}(\boldsymbol{h}_{\mathcal{G}})) \right)$, where $\boldsymbol{\mu}$ and $\boldsymbol{\Sigma}$ output the mean and log variance, respectively. How to use $\boldsymbol{z}$ is explained in Equation (1) and (2). The decoder consists of $\text{GNN}_{\text{pmol}}$, $\text{GNN}_{\text{motif}}$, $\text{NN}_{\text{query}}$, $\text{NN}_{\text{key}}$ and $\text{NN}_{\text{start}}$ that jointly work to generate molecules.

The overall training objective function is defined as:

$$\mathbb{E}_{\mathcal{G} \sim \mathcal{D}} \left[ \mathcal{L}(\mathcal{G}) \right] = \mathbb{E}_{\mathcal{G} \sim \mathcal{D}} \left[ \mathcal{L}_{rec}(\mathcal{G}) + \beta_{prior} \cdot \mathcal{L}_{prior}(\mathcal{G}) + \beta_{prop} \cdot \mathcal{L}_{prop}(\mathcal{G}) \right]. \tag{3}$$

In Equation (3): (1) $\mathcal{L}_{rec}(\mathcal{G})$ is the reconstruction loss as that in a standard VAE. It uses cross entropy loss to evaluate the likelihood of the reconstructed graph compared with the input $\mathcal{G}$. (2) The loss

---

[3] During inference, the motif representations can be calculated offline to avoid additional computation time.

[4] MiCaM can be naturally paired with other paradigms such as GAN or RL, which we plan to explore in future works.

$\mathcal{L}_{prior}(\mathcal{G}) = D_{\mathrm{KL}}(q(\cdot|\mathcal{G})\|\mathcal{N}(\mathbf{0}, \mathbf{I}))$ is used to regularize the posterior distribution in the latent space. (3) Following Maziarz et al. (2021), we add a property prediction loss $\mathcal{L}_{prop}(\mathcal{G})$ to ensure the continuity of the latent space with respect to some simple molecule properties. Specifically, we build another network $\mathrm{NN}_{prop}$ to predict the properties from the latent vector $\boldsymbol{z}$. (4) $\beta_{prior}$ and $\beta_{prop}$ are hyperparameters to be determined according to validation performances.

Here we emphasize two useful details. (1) Since MiCaM is an iterative method, in the training procedure, we need to determine the orders of motifs to be processed. The orders of the intermediate motifs (i.e., $t \geq 1$) are determined by the queue $\mathcal{Q}$ introduced in Section 2.2. The only remaining item is the first motif, and we choose the motif with the largest number of atoms as the first one. The intuition is that the largest motif mostly reflects the molecule's properties. The generation order is then determined according to Algorithm 2. (2) In the training procedure, we provide supervisions for every individual generation steps. We implement the reconstruction loss $\mathcal{L}_{rec}(\mathcal{G})$ by viewing the generation steps as parallel classification tasks. In practice, as the vocabulary size is large due to various possible connections, $\mathcal{L}_{rec}(\mathcal{G})$ is costly to compute. To tackle this problem, we subsample the vocabulary and modify $\mathcal{L}_{rec}(\mathcal{G})$ via contrastive learning (He et al., 2020). More details can be found in Appendix A.4.

## 2.4 DISCUSSION AND RELATED WORK

**Molecular Generation** A plethora of existing generative models are available and they fall into two categories: (1) string-based models (Kusner et al., 2017; Gómez-Bombarelli et al., 2018; Sanchez-Lengeling & Aspuru-Guzik, 2018; Segler et al., 2018), which rely on string representations of molecules such as SMILES (Weininger, 1988) and do not utilize the structual information of molecules, (2) and graph-based models (Liu et al., 2018; Guo et al., 2021) that are naturally based on molecule graphs. Graph-based approaches mainly include models that generate molecular graphs (1) atom-by-atom (Li et al., 2018; Mercado et al., 2021), and (2) fragment-by-fragment (Kong et al., 2021; Maziarz et al., 2021; Zhang et al., 2021; Guo et al., 2021). This work is mainly related to fragment-based methods.

**Motif Mining** Our motif mining algorithm is inspired by Byte Pair Encoding (BPE) (Gage, 1994), which is widely adapted in natural language processing (NLP) to tokenize words into subwords (Sennrich et al., 2015). Compared with BPE in NLP, molecules have much more complex structures due to different connections and bond types, which we solve by building the merging graphs. Another related class of algorithms are Frequent Subgraph Mining (FSM) algorithms (Kuramochi & Karypis, 2001; Jiang et al., 2013), which also aim to mine frequent motifs from graphs. However, these algorithms do not provide a "tokenizer" to fragmentize an arbitrary molecule into disjoint fragments, like what is done by BPE. Thus we cannot directly apply them in molecular generation tasks. Kong et al. (2021) also try to mine motifs, but they do not incorporate the connection information into the motif vocabulary and they apply a totally different generation procedure, which are important to the performance (see Appendix B.3). See Appendix D for more discussions.

**Motif Representation** Different from many prior works that view motifs as discrete tokens, we represent all the motifs in the motif vocabulary as graphs, and we apply the $\mathrm{GNN}_{\mathrm{motif}}$ to obtain the representations of motifs. This novel approach has three advantages. (1) The $\mathrm{GNN}_{\mathrm{motif}}$ obtains similar representations for similar motifs, which thus maintains the graph structure information of the motifs (see Appendix C.3). (2) The $\mathrm{GNN}_{\mathrm{motif}}$, combined with contrastive learning, can handle large size of motif vocabulary in training, which allows us to construct a large motif vocabulary (see Section 2.3 and Apppendix A.4). (3) The model can be easily transferred to another motif vocabulary. Thus we can jointly tune the motif vocabulary and the network parameters on a new dataset, improving the capability of the model to fit new data (see Section 3.2).

## 3 EXPERIMENTS

## 3.1 DISTRIBUTIONAL LEARNING RESULTS

To demonstrate the effectiveness of MiCaM, we test it on the benchmarks from GuacaMol (Brown et al., 2019), a commonly used evaluation framework to assess *de novo* molecular generation mod-

Table 1: Distributional results on QM9, ZINC, and GuacaMol. The higher the better for all metrics. The results of JT-VAE, GCPN and GP-VAE are from Kong et al. (2021). For MoLeR, we use the released code from Maziarz et al. (2021) with no changes.

| Dadaset | Model | Validity | Uniqueness | Novelty | KL Div | FCD |
|---------|-------|----------|------------|---------|--------|-----|
| QM9 | JT-VAE | **1.0** | 0.549 | 0.386 | 0.891 | 0.588 |
| | GCPN | **1.0** | 0.533 | 0.320 | 0.552 | 0.174 |
| | GP-VAE | **1.0** | 0.673 | **0.523** | 0.921 | 0.659 |
| | MoLeR | **1.0** | **0.940** | 0.355 | 0.969 | 0.931 |
| | MiCaM (Ours) | **1.0** | 0.932 | 0.493 | **0.980** | **0.945** |
| ZINC | JT-VAE | **1.0** | 0.988 | 0.988 | 0.882 | 0.263 |
| | GCPN | **1.0** | 0.982 | 0.982 | 0.456 | 0.003 |
| | GP-VAE | **1.0** | 0.997 | **0.997** | 0.850 | 0.318 |
| | MoLeR | **1.0** | 0.996 | 0.993 | 0.984 | 0.721 |
| | MiCaM (Ours) | **1.0** | **0.998** | **0.997** | **0.988** | **0.791** |
| GuacaMol | MoLeR | **1.0** | **1.000** | **0.991** | 0.964 | 0.625 |
| | MiCaM (Ours) | **1.0** | 0.994 | 0.986 | **0.989** | **0.731** |

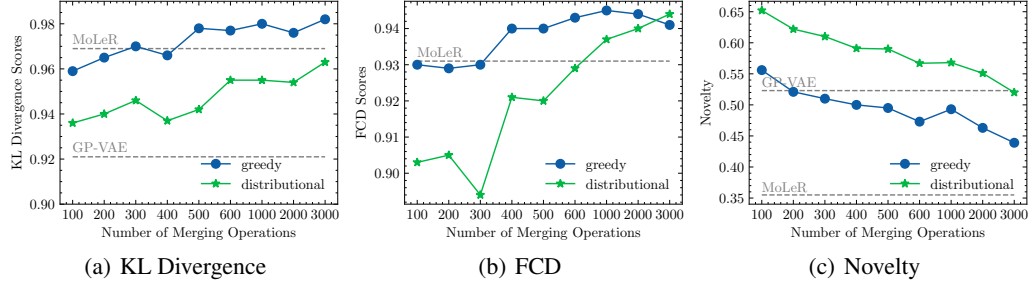

(a) KL Divergence     (b) FCD     (c) Novelty

Figure 4: KL Divergence and FCD scores (higher is better) for different numbers of merging operations and different choices of generation modes.

els.[5] We first consider the distribution learning benchmarks, which assess how well the models learn to generate novel molecules that resemble the distribution of the training set.

**Experimental Setup** Following Brown et al. (2019), we consider five metrics for distribution learning: validity, uniqueness, novelty, KL divergence (KL Div) and Fréchet ChemNet Distance (FCD). The first three metrics measure if the model can generate chemically valid, unique, and novel candidate molecules, while last two are designed to measure the distributional similarity between the generated molecules and the training set. For the KL Div benchmark, we compare the probability distributions of a variety physicochemial descriptors. A higher KL Div score, i.e., lower KL divergences for the descriptors, means that the generated molecules resemble the training set in terms of these descriptors. The FCD is calculated from the hidden representations of molecules in a neural network called ChemNet (Preuer et al., 2018), which can capture important chemiacal and biological features of molecules. A higher FCD score, i.e., a lower Fréchet ChemNet Distance means that the generated molecules have similar chemical and biological properties to those from the training set.

We evaluate our method on three datasets: QM9 (Ruddigkeit et al., 2012), ZINC (Irwin et al., 2012), and GuacaMol (a post-processed ChEMBL (Mendez et al., 2019) dataset proposed by Brown et al. (2019)). These three datasets cover different molecule complexities and different data sizes. The results across them demonstrate the capability of our model to handle different kinds of data.

---

[5]The code of MiCaM is available at https://github.com/MIRALab-USTC/AI4Sci-MiCaM.

Table 2: Goal directed generation results on five GuacaMol benchmarks. The higher the better for all metrics. The results of other baselines are from Brown et al. (2019) and Ahn et al. (2020). The benchmarks are: a. Celecoxib Rediscovery; b. Aripiprazole Similarity; c. $C_{11}H_{24}$ Isomers; d. Ranolazine MPO; and e. Sitagliptin MPO.

| Benchmark | Dataset | SMILES GA | Graph MCTS | Graph GA | SMILES LSTM | MSO | MiCaM (Ours) |
|-----------|---------|-----------|------------|----------|-------------|-----|--------------|
| a. | 0.505 | 0.732 | 0.355 | **1.000** | **1.000** | **1.000** | **1.000** |
| b. | 0.595 | 0.834 | 0.380 | **1.000** | **1.000** | **1.000** | **1.000** |
| c. | 0.684 | 0.829 | 0.410 | 0.971 | 0.993 | 0.997 | **0.999** |
| d. | 0.792 | 0.881 | 0.616 | 0.920 | 0.855 | 0.931 | **0.932** |
| e. | 0.509 | 0.689 | 0.458 | 0.891 | 0.545 | 0.868 | **0.914** |

**Quantitative Results** Table 1 presents our experimental results of distribution learning tasks. We compare our model with several state-of-the-art models: JT-VAE (Jin et al., 2018), GCPN (You et al., 2018), GP-VAE (Kong et al., 2021), and MoLeR (Maziarz et al., 2021). Since all the models are graph-based, they obtain 100% validity by introducing chemical valence check during generation. We can see that MiCaM achieves the best performances on the KL Divergence and FCD scores on all the three datasets, which demonstrates that it can well resemble the distributions of training sets. Meanwhile it keeps high uniqueness and novelty, comparable to previous best results. In this experiment, we set the number of merging operations to be 1000 for QM9, due to the results in Figure 4. For ZINC and GuacaMol, we simply set the number to be 500 and find that MiCaM has achieved performances that outperform all the baselines. This indicates that existing methods tend to perform well on the sets of relatively simple molecules such as QM9, while MiCaM performs well on datasets with variant complexity. We further visualize the distributions in Figure 7 of Appendix B.2.

**Number of Merging Operations** We conduct experiments on different choices of the number of merging operations. Figure 4 presents the experimental results on QM9. It shows that FCD score and KL Divergence score, which measure the similarity between the generated molecules and the training set, increase as the number of merging operations grows. Meanwhile, the novelty decreases as the number grows. Intuitively, the more merging operations we use for motif vocabulary construction, the larger motifs will be contained in the vocabulary, and thus will induce more structural information from the training set. We can achieve a trade-off between the similarity and novelty by controlling the number of merging operations. Empirically, a medium number (about 500) of operations is enough to achieve a high similarity.

**Generation Modes** We also compare two different generation modes, i.e., the greedy mode and the distributional mode. With the greedy mode, the model always picks the motif or the connection site with the highest probability. While the distributional mode allows picking motifs or connection sites according to a distribution. The results show that the greedy mode leads to a little higher KL Divergence and FCD scores, while the distributional mode leads to a higher novelty.

## 3.2 GOAL DIRECTED GENERATION RESULTS

We further demonstrate the capability of our model to generate molecules with wanted properties. In such goal directed generation tasks, we aim to generate molecules that have high scores which are predefined by rules.

**Iteratively Tuning** We combine MiCaM with iterative target augmentation (ITA) (Yang et al., 2020) and generate optimized molecules by iteratively generating new molecules and tuning the model on molecules with highest scores. Specifically, we first pick out $N$ molecules with top scores from the GuacaMol dataset and store them in a training buffer. Iteratively, we tune our model on the training buffer and generate new molecules. In each iteration, we update the training buffer to store the top $N$ molecules that are either newly generated or from the training buffer in the last iteration. In order to accelerate the model to explore the latent space, we pair MiCaM with Molecular Swarm Optimization (MSO) (Winter et al., 2019) in each iteration to generate new molecules.

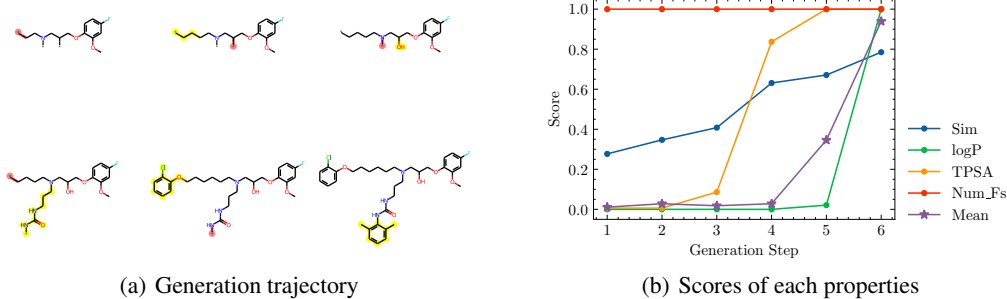

| (a) Generation trajectory | (b) Scores of each properties |

Figure 5: A generation trajectory for Ranolazine MPO benchmark. In each generation step, the query connection is marked in red, and the newly added motif is marked in yellow.

A novelty of our approach is that when tuning on a new dataset, we jointly update the motif vocabulary and the network parameters. We can do this because we apply a $\text{GNN}_{\text{motif}}$ to obtain motif representations, which can be transferred to a newly built motif vocabulary. The representations of the new motifs are calculated by $\text{GNN}_{\text{motif}}$, and we then optimize the network parameters using both new data and newly constructed motif vocabulary.

**Experimental Setup**  We test MiCaM on several goal directed generation tasks from GuacaMol benchmarks. Specifically, we consider four different categories of tasks: a Rediscovery task (Celecoxib Rediscovery), a Similarity task (Aripiprazole Similarity), an Isomers task ($C_{11}H_{24}$ Isomers), and two multi-property objective (MPO) tasks (Ranolazine MPO and Sitagliptin MPO). We compare MiCaM with several strong baselines.

**Quantitative Results**  For each benchmark, we run 7 iterations. In each iteration, we apply MSO to generate $80,000$ molecules, and store $10,000$ molecules with highest scores in the training buffer to tune MiCaM. We tune the model pretrained on the GuacaMol dataset and Table 2 presents the results. MiCaM achieves $1.0$ scores on some relatively easy benchmarks. It achieves high scores on several difficult benchmarks such MPO tasks, outperforming the baselines.

**Case Studies**  There are some domain-specific motifs that are beneficial to the target properties in different goal directed generation tasks, which are likely to be the pharmacophores in drug molecules. We conduct case studies to demonstrate the ability of MiCaM to mine such favorable motifs for domain-specific tasks.

In Figure 5 we present cases for the Ranolazine MPO benchmark, which tries to discover molecules similar to Ranolazine, a known drug molecule, but with additional requirements on some other properties. This benchmark calculates the geometric mean of four scores: Sim (the similarity between the molecule and Ranolazine), logP, TPSA, and Num_Fs (the number of fluorine atoms). We present a generation trajectory as well as the scores in each generation step. Due to the domain-specific motif vocabulary and the connection query mechanism, it requires only a few steps to generate such a complex molecule. Moreover, we can see that the scores increase as some key motifs are added to the molecule, which implies that the picked motifs are relevant to the target properties. See Figure 8 in Appendix B.4 for more case studies.

## 4  CONCLUSION

In this work, we proposed MiCaM, a novel model that generates molecules based on mined connection-aware motifs. Specifically, the contributions include (1) a data-driven algorithm to mine motifs by iteratively merging most frequent subgraph patterns and (2) a connection-aware generator for *de novo* molecular generation. It achieve state-of-the-art results on distribution learning tasks and on three different datasets. Combined with iterative target augmentation, it can learn domain-specific motifs related to some properties and performs well on goal directed benchmarks.

ACKNOWLEDGEMENT

The authors would like to thank all the anonymous reviewers for their insightful comments. This work was supported in part by National Nature Science Foundations of China grants U19B2026, U19B2044, 61836011, 62021001, and 61836006, and the Fundamental Research Funds for the Central Universities grant WK3490000004.

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

## A  IMPLEMENTATION DETAILS

### A.1  MOTIF MINING ALGORITHM

---

**Algorithm 1:** Connection-aware Motif Mining

---

**Input:** A set of molecule graphs $\mathcal{D} = \{\mathcal{G}_1, \mathcal{G}_2, \cdots, \mathcal{G}_{|\mathcal{D}|}\}$, the number $K$ of iterations.

**Output:** The merging operations $\{\mathcal{M}^{(k)}\}_{k=0}^{k-1}$ and the motif vocabulary Vocab $= \{\mathcal{F}^*\}$.

1 $\mathcal{D}_M^{(0)} \leftarrow \{\}$;           `// Merging-operation learning phase`
2 **for** $\mathcal{G}(\mathcal{V}, \mathcal{E}) \in \mathcal{D}$ **do**
3     $\mathcal{G}_M^{(0)}(\mathcal{V}_M^{(0)}, \mathcal{E}_M^{(0)}) \leftarrow \mathcal{G}(\mathcal{V}, \mathcal{E})$;
4     $\mathcal{D}_M^{(0)} \leftarrow \mathcal{D}_M^{(0)} \cup \{\mathcal{G}_M^{(0)}\}$;
5 **for** $k = 0$ *to* $K - 1$ **do**
6     Reset Count() to 0;
7     **for** $\mathcal{G}_M^{(k)}(\mathcal{V}_M^{(k)}, \mathcal{E}_M^{(k)}) \in \mathcal{D}_M^{(k)}$ **do**
8        **for** $(\mathcal{F}_i, \mathcal{F}_j) \in \mathcal{E}_M^{(k)}$ **do**
9           $\mathcal{M} \leftarrow \mathcal{F}_i \oplus \mathcal{F}_j$;
10           $\text{Count}(\mathcal{M}) \leftarrow \text{Count}(\mathcal{M}) + 1$;
11     $\mathcal{M}^{(k)} \leftarrow \arg\max \text{Count}(\mathcal{M})$;
12     $\mathcal{D}_M^{(k+1)} \leftarrow \{\}$;
13     **for** $\mathcal{G}_M^{(k)}(\mathcal{V}_M^{(k)}, \mathcal{E}_M^{(k)}) \in \mathcal{D}_M^{(k)}$ **do**
14        $\mathcal{G}_M^{(k+1)}(\mathcal{V}_M^{(k+1)}, \mathcal{E}_M^{(k+1)}) \leftarrow \mathcal{G}_M^{(k)}(\mathcal{V}_M^{(k)}, \mathcal{E}_M^{(k)})$;
15        **for** $(\mathcal{F}_i, \mathcal{F}_j) \in \mathcal{E}_M^{(k)}$ **do**
16           **if** $(\mathcal{F}_i, \mathcal{F}_j) \in \mathcal{E}_M^{(k+1)}$ && $\mathcal{F}_i \oplus \mathcal{F}_j == \mathcal{M}^{(k)}$ **then**
17             Merge $\mathcal{F}_i$ and $\mathcal{F}_j$ in $\mathcal{G}_M^{(k+1)}$;
18        $\mathcal{D}_M^{(k+1)} \leftarrow \mathcal{D}_M^{(k+1)} \cup \{\mathcal{G}_M^{(k+1)}\}$;
19 Vocab $\leftarrow \{\}$;           `// Motif-vocabulary construction phase`
20 **for** $\mathcal{G}(\mathcal{V}, \mathcal{E}) \in \mathcal{D}$ **do**
21     $\mathcal{G}_M^{(0)}(\mathcal{V}_M^{(0)}, \mathcal{E}_M^{(0)}) \leftarrow \mathcal{G}(\mathcal{V}, \mathcal{E})$;
22     **for** $k = 0$ *to* $K - 1$ **do**
23        $\mathcal{G}_M^{(k+1)}(\mathcal{V}_M^{(k+1)}, \mathcal{E}_M^{(k+1)}) \leftarrow \mathcal{G}_M^{(k)}(\mathcal{V}_M^{(k)}, \mathcal{E}_M^{(k)})$;
24        **for** $(\mathcal{F}_i, \mathcal{F}_j) \in \mathcal{E}_M^{(k)}$ **do**
25           **if** $(\mathcal{F}_i, \mathcal{F}_j) \in \mathcal{E}_M^{(k+1)}$ && $\mathcal{F}_i \oplus F_j == \mathcal{M}^{(k)}$ **then**
26             Merge $\mathcal{F}_i$ and $\mathcal{F}_j$ in $\mathcal{G}_M^{(k+1)}$;
27     Vocab $\leftarrow$ Vocab $\cup \{\mathcal{F}^* \text{ for } \mathcal{F} \in \mathcal{V}_M^{(K)}\}$;

---

Our connection-aware motif mining algorithm is in Algorithm 1. In the merging graph $\mathcal{G}_M(\mathcal{V}_M, \mathcal{E}_M)$, each node $\mathcal{F} \in \mathcal{V}_M$ represents a fragment $\mathcal{F}(\hat{\mathcal{V}}, \hat{\mathcal{E}})$ of the molecule graph $\mathcal{G}(\mathcal{V}, \mathcal{E})$, where $\hat{\mathcal{V}} \subset \mathcal{V}$ and $\hat{\mathcal{E}} \subset \mathcal{E}$ are atoms and bonds in $\mathcal{F}$, respectively. The edge set $\mathcal{E}_M$ is defined by: for any two fragments $F_i(\mathcal{V}_i, \mathcal{E}_i), F_j(\mathcal{V}_j, \mathcal{E}_j) \in \mathcal{V}_M$, $(\mathcal{F}_i, \mathcal{F}_j) \in \mathcal{E}_M \iff \exists a \in \mathcal{V}_i, b \in \mathcal{V}_j, (a, b) \in \mathcal{E}$. The merging operation "$\oplus$" is defined to create $\mathcal{F}_{ij}(\mathcal{V}_{ij}, \mathcal{E}_{ij}) = \mathcal{F}_i \oplus \mathcal{F}_j$ by merging two fragments $\mathcal{F}_i(\mathcal{V}_i, \mathcal{E}_i)$ and $\mathcal{F}_j(\mathcal{V}_j, \mathcal{E}_j)$. Formally,

$$\mathcal{V}_{ij} = \mathcal{V}_i \cup \mathcal{V}_j, \quad \mathcal{E}_{ij} = \mathcal{E}_i \cup \mathcal{E}_j \cup \{(a, b) \in \mathcal{E} | a \in \mathcal{V}_i, b \in \mathcal{V}_j\},$$

which means the new fragment $\mathcal{F}_{ij}$ contains all nodes and edges from $\mathcal{F}_i, \mathcal{F}_j$ and the edges between them. Notice that when we traverse the edges of graphs, we always follow the orders determined by RDKit. In the motif-vocabulary construction phase, we disconnect bonds between different fragments and add "*" atoms to create connection-aware motifs. Specifically, for each $\mathcal{F}(\hat{\mathcal{V}}, \hat{\mathcal{E}}) \in \mathcal{V}_M$, we define its corresponding connection-aware motif $\mathcal{F}^*(\hat{\mathcal{V}}^*, \hat{\mathcal{E}}^*)$ as

$$\hat{\mathcal{V}}^* = \hat{\mathcal{V}} \cup \{a^* | a \in \mathcal{V}, \exists b \in \hat{\mathcal{V}}, (a, b) \in \mathcal{E}\},$$

$$\hat{\mathcal{E}}^* = \{(a, b) \in \mathcal{E} | a \in \hat{\mathcal{V}}^*, b \in \hat{\mathcal{V}}^*\},$$

where "$a^*$" denotes that we change the label of the atom $a$ to "$*$". The symbol "$*$" can be seen as a dummy atom or a connection site, which indicates that the bond is non-terminal and we will grow the molecule here.

**Efficiency**   Figure 6 presents the time costs of learning operations from QM9, which demonstrates that our algorithm is fast, and the time cost of each iteration decreases rapidly as the motif frequency decreases.

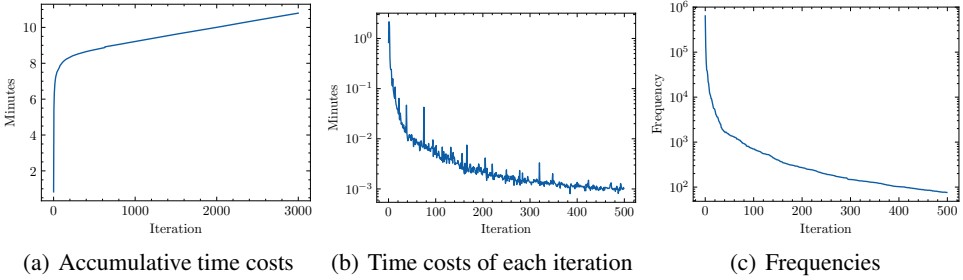

(a) Accumulative time costs  (b) Time costs of each iteration  (c) Frequencies

Figure 6: Time costs of learning merging operations on QM9. (a) and (b) show the accumulative time costs and time costs of each iteration, respectively. (c) shows the frequencies of the learnt merging operations.

**Sequential Merging Operations**   The merging operations are useful as they work as a "tokenizer" that can be applied to fragmentize an arbitrary molecule, which is unavailable for other methods like Frequent Subgraph Mining. Another choice of using the merging operations is not applying them sequentially, but traversing the molecule edges to find if there are patterns that appear in the learnt frequent motifs. However, this is sub-optimal as it does not work reasonably on any arbitrary molecule outside the dataset to learn the operations. For example, consider a trivial dataset $\mathcal{D} = \{CC, CN, CNN, CN=O, CC=O\}$. When we run two iterations, the learnt merging operations are $\{CN, CC\}$. Then we use them to fragmentize a new molecule "CCN". If we apply the merging operations sequentially, the molecule will be decomposed into $\{C, CN\}$. However, if we traverse the molecule edges to find the patterns, the molecule will be decomposed into $\{CC, N\}$, which is sub-optimal because "CN" appears with a higher frequency in the dataset.

## A.2   GENERATING PROCEDURE

Algorithm 2 presents the pseudo code of our generation procedure. For ease of reference, let $\mathcal{C}_{\mathcal{G}}$ denote the connection sites from a graph $\mathcal{G}$ (either a partial molecule or a motif), and let $\mathcal{C}_{\text{Vocab}} = \cup_{\mathcal{F}^* \in \text{Vocab}} \mathcal{C}_{\mathcal{F}^*}$ be the set of all connection sites from the motif vocabulary.

## A.3   NETWORKS

The backbone of MiCaM is VAE. The encoder is $\text{GNN}_{\text{mol}}$, followed by three MLPs: $\boldsymbol{\mu}$ and $\boldsymbol{\Sigma}$ for resampling, and $\text{NN}_{\text{prop}}$ for property prediction. The decoder consists of $\text{GNN}_{\text{pmol}}$, $\text{GNN}_{\text{motif}}$, which are GNNs, and $\text{NN}_{\text{start}}$, $\text{NN}_{\text{query}}$, $\text{NN}_{\text{key}}$, which are MLPs. All MLPs have 3 layers and use ReLU as the activation function. The latent size and the hidden size are both $256$.

We employ GINE (Hu et al., 2019) as the GNN structures, and in each GNN layer we employ a 3-layer MLP for messaage aggregation. For all GNNs, we use five atom-level features as inputs: atom symbol, is_aromatic, formal charge, num_explicit_Hs, num_implicit_Hs. The features are embedded with the dimension $192, 16, 16, 16, 16$, respectively, and thus the node embedding size is $256$. Four edges, we consider four types of bonds (single bonds, double bonds, triple bonds and aromatic bond), and the embedding size is $256$. $\text{GNN}_{\text{mol}}$ and $\text{GNN}_{\text{pmol}}$ have $15$ layers and $\text{GNN}_{\text{motif}}$ has $6$ layers. You can see our released code for more details.

---

**Algorithm 2:** Generating a molecule

---

**Input:** A connection-aware motif vocabulary Vocab $= \{\mathcal{F}^*\}$. A latent vector $z$.
**Output:** A molecule graph $\mathcal{G}$.

**1** $\mathcal{C}_{\text{Vocab}} \leftarrow \bigcup_{\mathcal{F}^* \in \text{Vocab}} \mathcal{C}_{\mathcal{F}^*}$;
**2** Calculate $P_{\mathcal{F}^*}$ by Equation 2 and sample $\mathcal{F}^* \sim P_{\mathcal{F}^*}$;
**3** $\mathcal{G} \leftarrow \mathcal{F}^*, \mathcal{Q} \leftarrow \emptyset$;                   // Pick the starting motif
**4** **for** $v \in \mathcal{C}_{\mathcal{F}^*}$ **do**
**5**    |  $\mathcal{Q}.put(v)$;
**6** **while** $\mathcal{Q} \neq \emptyset$ **do**                   // Connection querying step
**7**    |  $v_t \leftarrow \mathcal{Q}.get()$;
**8**    |  $\mathcal{C} \leftarrow \mathcal{C}_{\text{Vocab}} \cup \mathcal{C}_{\mathcal{G}} \setminus \{v_t\}$;
**9**    |  Calculate $P_v$ over $\mathcal{C}$ by Equation 1 and sample $v \sim P_v$;
**10**    |  **if** $v \in \mathcal{C}_{Vocab}$ and $v$ in $\mathcal{F}^* \in Vocab$ **then**
**11**    |    |  $\mathcal{G} \leftarrow \mathcal{G}.\text{AddMotif}(\mathcal{F}^*)$;        // Connecting a new motif
**12**    |    |  $\mathcal{G} \leftarrow \mathcal{G}.\text{Merge}(v_t, v)$;
**13**    |    |  **for** $v' \in \mathcal{C}_{\mathcal{F}^*} \setminus \{v\}$ **do**
**14**    |    |    |  $\mathcal{Q}.put(v')$;
**15**    |  **else**
**16**    |    |  Assert $v \in \mathcal{C}_{\mathcal{G}} \setminus \{v_t\}$;
**17**    |    |  $\mathcal{G} \leftarrow \mathcal{G}.\text{Merge}(v_t, v)$;            // Cyclizing itself
**18**    |    |  $\mathcal{Q} \leftarrow \mathcal{Q} \setminus \{v\}$

---

### A.4 EXPERIMENT DETAILS

**Learning Merging Operations** For QM9, we apply 1000 merging operations due to the comparative results in 4. For ZINC and GuacaMol, we simply use 500 merging operations without elaborate searching, and find that it has achieved good results. Due to the large size of GuacaMol dataset, we randomly sample $100,000$ molecules from it to learn merging operations, and then apply these merging operations on all molecules to obtain the motif vocabulary.

**Training MiCaM** We preprocess all the molecules to provide supervision signals for decoder to rebuild the molecules. We provide the true indices of picked connections in every steps so that the model can learn the ground truth. In each optimization step, we update the network parameters by optimizing the loss function on a batch $\mathcal{B}$ of molecules:

$$\mathcal{L}_{\mathcal{B}} = \mathbb{E}_{\mathcal{G} \sim \mathcal{B}} \left[ \beta_{prior} \cdot \mathcal{L}_{prior}(\mathcal{G}) + \mathcal{L}_{rec}(\mathcal{G}) + \beta_{prop} \cdot \mathcal{L}_{prop}(\mathcal{G}) \right].$$

Specifically, the reconstruction loss $\mathcal{L}_{rec}$ is written as a sum over the negative log probabilities of the partial graphs $\mathcal{G}_t$ at each step $t$, conditioned on $z$ and the last steps:

$$\mathcal{L}_{rec}(\mathcal{G}) = -\mathbb{E}_{z \sim q(\cdot|\mathcal{G})} \left[ \log p(\mathcal{G}_0|z) + \sum_t \log p(\mathcal{G}_{t+1}|z, \mathcal{G}_t) \right]$$

$$= -\mathbb{E}_{z \sim q(\cdot|\mathcal{G})} \left[ \log p(\mathcal{F}_0^*|z) + \sum_t \log p(u_t|z, \mathcal{G}_t, v_t) \right],$$

where $\mathcal{F}_0^*$, $v_t$ and $u_t$ are the first motif, the focused query connection site and the picked connection at the $t^{\text{th}}$ step, respectively. In practice, since the motif vocabulary is large due to different connections, we modify the loss via contrastive learning for efficient training:

$$\log p(\mathcal{F}_0^*|z) \leftarrow \log \frac{\exp(\text{NN}_{\text{start}}(z) \cdot \text{NN}_{\text{key}}(h_{\mathcal{F}_0^*}))}{\sum_{\mathcal{F}^* \in \mathcal{I}_{\mathcal{F}^*}} \exp(\text{NN}_{\text{start}}(z) \cdot \text{NN}_{\text{key}}(h_{\mathcal{F}^*}))},$$

$$\log p(u_t|z, \mathcal{G}_t, v_t) \leftarrow \log \frac{\exp(\text{NN}_{\text{query}}([z, h_{\mathcal{G}_t}, h_{v_t}]) \cdot \text{NN}_{\text{key}}(h_{u_t}))}{\sum_{v \in \mathcal{I}_v} \exp(\text{NN}_{\text{query}}([z, h_{\mathcal{G}_t}, h_{v_t}]) \cdot \text{NN}_{\text{key}}(h_v))},$$

where $\mathcal{I}_{\mathcal{F}^*}$ and $\mathcal{I}_v$ are the sets of motifs and connections, respectively, containing a positive sample and negative samples from the batch $\mathcal{B}$.

We find that a proper choice of $\beta_{prior}$ is essentially important to the performances on distribution learning benchmarks, especially for FCD scores. For QM9, we use a short warm-up ($3,000$ steps), and use a long sigmoid schedule ($400,000$ steps) (Bowman et al., 2015) to let $\beta_{prior}$ to reach $0.4$.

For property prediction, we predict four simple properties of molecules, including molecular weight, synthetic accessibility (SA) score, octanol-water partition coefficient (logP) and quantitative estimate of drug-likeness (QED). The target values are computed using the RDKit library. Empirically, for distribution learning benchmarks, a small $\beta_{prop}$ (about $0.3$) is beneficial.

## A.5   VALIDITY CHECK

We conduct a validity check during generation to avoid the model generating invalid aromatic rings (e.g., merging two "*:c:c:c:c:*"s into "c1ccccccc1"). Specifically, when the model tries to generate such an invalid aromatic ring, we simply remove the aromaticity of this ring so that the molecule is still valid (e.g., "c1ccccccc1" will be replaced with "C1CCCCCCC1"). Without this chemical validity check, the validity rates on QM9, ZINC, and GuacaMol are $99.68\%$, $98.6\%$, and $98.28\%$, respectively. The high validity rates indicate that MiCaM learns to generate valid aromatic rings.

# B   ADDITIONAL RESULTS

## B.1   EFFICIENCY

Thanks to the mined motifs and the connection-aware decoder, MiCaM is very efficient. We measure the training and sampling speed on a single GeForce RTX 3090. For training, it trains on $325.7$ molecules per second. For inference, it generates $54.4$ molecules per second.

## B.2   DISTRIBUTION VISUALIZATION

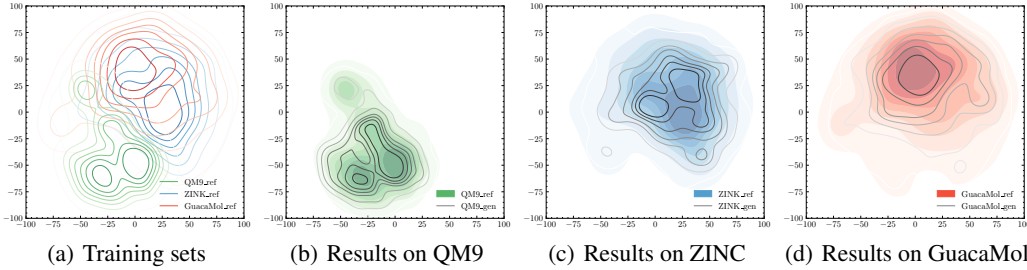

| (a) Training sets | (b) Results on QM9 | (c) Results on ZINC | (d) Results on GuacaMol |

Figure 7: Visualization of the probability distributions of training sets (QM9, ZINC and GuacaMol) and the generated molecules. The postfix "_ref" means reference, i.e., the training sets (shown in green, blue and red, respectively), and the postfix "_gen" means the sets of molecules generated by our model (shown in grey). We obtain the representations of molecules by calculating their molecular fingerprints, and we then apply t-SNE dimensionality reduction for visualization. The curves represent the contour lines of the probability distributions of the datasets. (a) shows the distributional shift over the three training sets. (b), (c) and (d) demonstrate that our generative model can properly fit the different distributions respectively.

We visualize the distributional results on the three datasets in Figure 7. Specifically, we first calculate the Morgan fingerprints of all molecules. Morgan fingerprint has been used for a long time in drug discovery, as it can represents the structual information of molecules (Rogers & Hahn, 2010). For visualization, we apply the t-distributed stochastic neighbor embedding (t-SNE) algorithm (Van der Maaten & Hinton, 2008)—a nonlinear dimensionality reduction technique to keep the similar high-dimensional vectors close in lower-dimensional space—to represent molecules in the two-dimensional plane, and then plot the distributions of molecules.

B.3 ABLATION STUDY

**Trade-off Among Metrics**   As KL Divergence and FCD negatively correlate with Uniqueness and Novelty, there is a trade-off among the metrics in distribution learning tasks. We can achieve this trade-off via some hyperparameters. For example, on QM9, if we conduct 500 merging operations, and use distributional mode (sampling from top 5 choices) for sampling, MiCaM achieves higher uniqueness and novelty and outperforms MoLeR in terms of all the metrics. The results are in Table 3.

Table 3: MiCaM can achieve a trade-off among the distribution learning metrics. With 1000 merging operations and greedy mode, MiCaM significantly outperforms MoLeR in terms of KL Divergence and FCD. While with 500 merging operations and distribution mode, MiCaM outperforms MoLeR in terms of all the metrics.

| Model | Validity | Uniqueness | Novelty | KL Div | FCD |
|---|---|---|---|---|---|
| MoLeR | **1.0** | 0.940 | 0.355 | 0.969 | 0.931 |
| MiCaM-100-greedy | **1.0** | 0.932 | 0.493 | **0.980** | **0.945** |
| MiCaM-500-distr | **1.0** | **0.941** | **0.495** | 0.978 | 0.940 |

**Motif Vocabulary**   We conduct two more experiments on QM9 to verify the effect of the motif vocabulary. Specifically, we use the generation procedure of MiCaM, but replace the motif vocabulary with the vocabularies in MoLeR (Maziarz et al., 2021) and MGSSL (Zhang et al., 2021), respectively. For a fair comparison, we preserve the connection information (i.e., the "*"s) in the two new vocabularies. We name the two models MiCaM-moler and MiCaM-brics, respectively, as MGSSL applies BRICS (Degen et al., 2008) with further decomposition. The results are in Table 4.

Table 4: Ablation studies on different motif vocabularies on QM9.

| Model | Validity | Uniqueness | Novelty | KL Div | FCD |
|---|---|---|---|---|---|
| MiCaM-moler | **1.0** | 0.926 | 0.468 | 0.973 | 0.934 |
| MiCaM-brics | **1.0** | 0.927 | 0.485 | 0.978 | 0.938 |
| MiCaM | **1.0** | **0.932** | **0.493** | **0.980** | **0.945** |

**Generating Procedure**   Besides the molecule fragmentation strategy, two components contribute to the performance of MiCaM. First is the connection information preserved in the motif vocabulary and corresponding connection-aware decoder. Second is the $GNN_{motif}$, which captures the graph structures of motifs, and allows efficient training on a large motif vocabulary via contrastive learning. We conduct ablation studies to demonstrate the importance of the two components. Specifically, we implement three different versions of MiCaM. MiCaM-v1 does not apply $NN_{motif}$ and does not use connection information for generation. In each step, it first picks up the motif (without connection information) by viewing them as discrete tokens, and then determines the connecting points and bonds. MiCaM-v2 employs the $NN_{motif}$ to pick up motifs, but does not directly query the connection sites. The results demonstrate that leveraging connection information and employing the $GNN_{motif}$ actually bring performance improvements.

Table 5: Ablation studies on different versions of MiCaM. The results are from a set of $100,000$ molecules randomly sampled from GuacaMol.

| Model | Validity | Uniqueness | Novelty | KL Div | FCD |
|---|---|---|---|---|---|
| MiCaM-v1 | **1.0** | **1.000** | 0.993 | 0.926 | 0.685 |
| MiCaM-v2 | **1.0** | 0.989 | 0.989 | **0.965** | 0.617 |
| MiCaM | **1.0** | 0.998 | **0.995** | 0.957 | **0.754** |

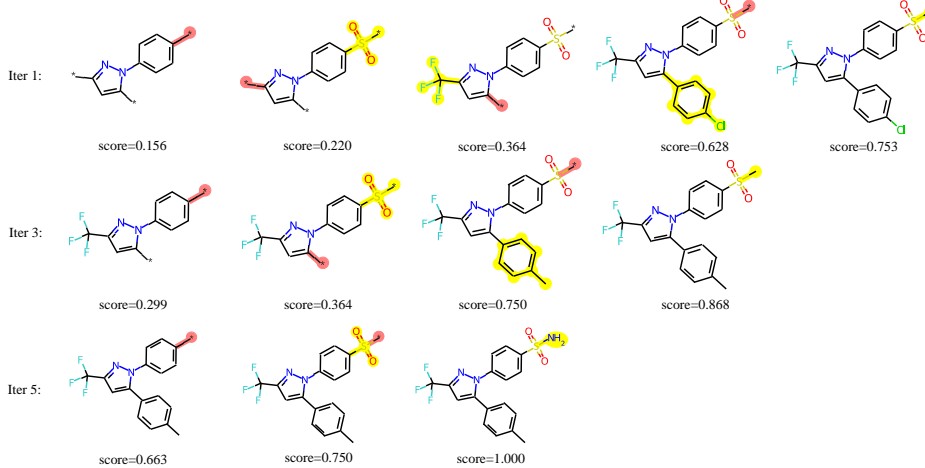

Figure 8: Generation trajectories for Celecoxib Rediscovery. We show the trajectories of the best molecules in three different iterations. In each generation step, the query connection is marked in red, and the newly added motif is marked in yellow.

## B.4 CASE STUDIES

Figure 8 presents cases for the Celecoxib Rediscovery task, which aims to discover molecules similar to Celecoxid, a known drug molecule. Specifically, we present the trajectories to generate molecules with the highest scores in the 1st, 3rd and 5th iterations. As the number of iteration increases, the motifs learnt by MiCaM tend to be more specific and more adaptive to the target, leading to the model to generate molecules with higher scores while costing fewer generation steps.

## B.5 GENERATED MOLECULES

Some examples of the generated molecules are in Figure 9. For further comparison, we visualize the probability distributions of GuacaMol, the molecules generated by MiCaM and MoLeR, respectively, in Figure 10. We can see that MiCaM fits the reference distribution better than MoLeR. Moreover, from the visualization, we find that the outermost contour line of MiCaM covers more area than MoLeR and fits that of the reference data better. This indicates that some reasonable chemical spaces are explored more by MiCaM than MoLeR. We then find that such cases include molecules with large rings or complex ring systems. See Figure 11 for some concrete examples.

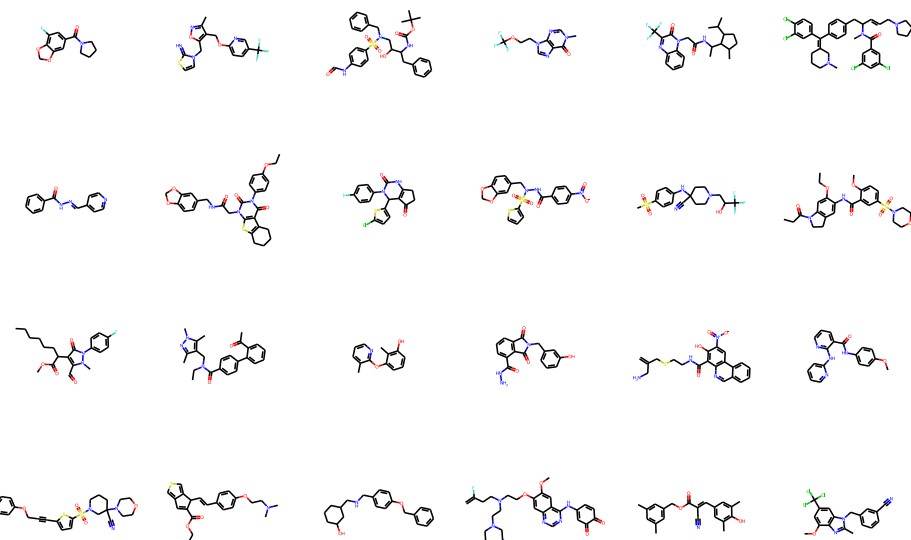

Figure 9: Samples from molecules randomly generated by a MiCaM model, trained on GuacaMol.

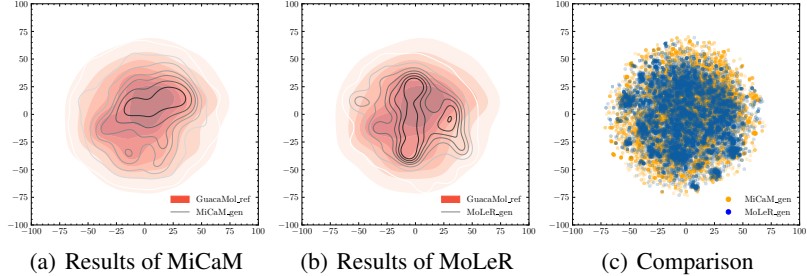

(a) Results of MiCaM      (b) Results of MoLeR      (c) Comparison

Figure 10: Visualization of the distributions of the molecule sets. We obtain the representations of molecules by calculating their molecular fingerprints, and we then apply t-SNE dimensionality reduction for visualization. (a) and (b) visualize the contour lines of the probability distributions of the molecule sets. Red represents the GucaMol datasets, while gray represents the generated molecules. (c) shows the samples from MiCaM and MoLeR in orange and blue, respectively. Each point represents a molecule.

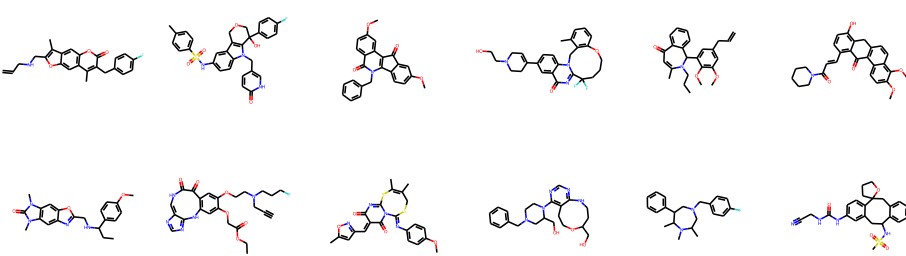

Figure 11: Molecules with large rings or complex ring systems generated by MiCaM, which are not likely from MoLeR.

# C  MOTIF VOCABULARY

## C.1  MERGING OPERATIONS

We present the learnt merging operations from QM9 and GuacaMol in Figure 12. The merging operations can efficiently merge molecules into a few disjoint units. For statistics, each molecule in GuacaMol has 27.899 atoms on average. After applying 500 merging operations, each of they is represented as only 8.499 subgraphs on average. After applying 1000 merging operations, the number is 8.282. As a comparison, MoLeR decomposes each molecule into 9.667 fragments on average, when taking 4096 as the vocabulary size.

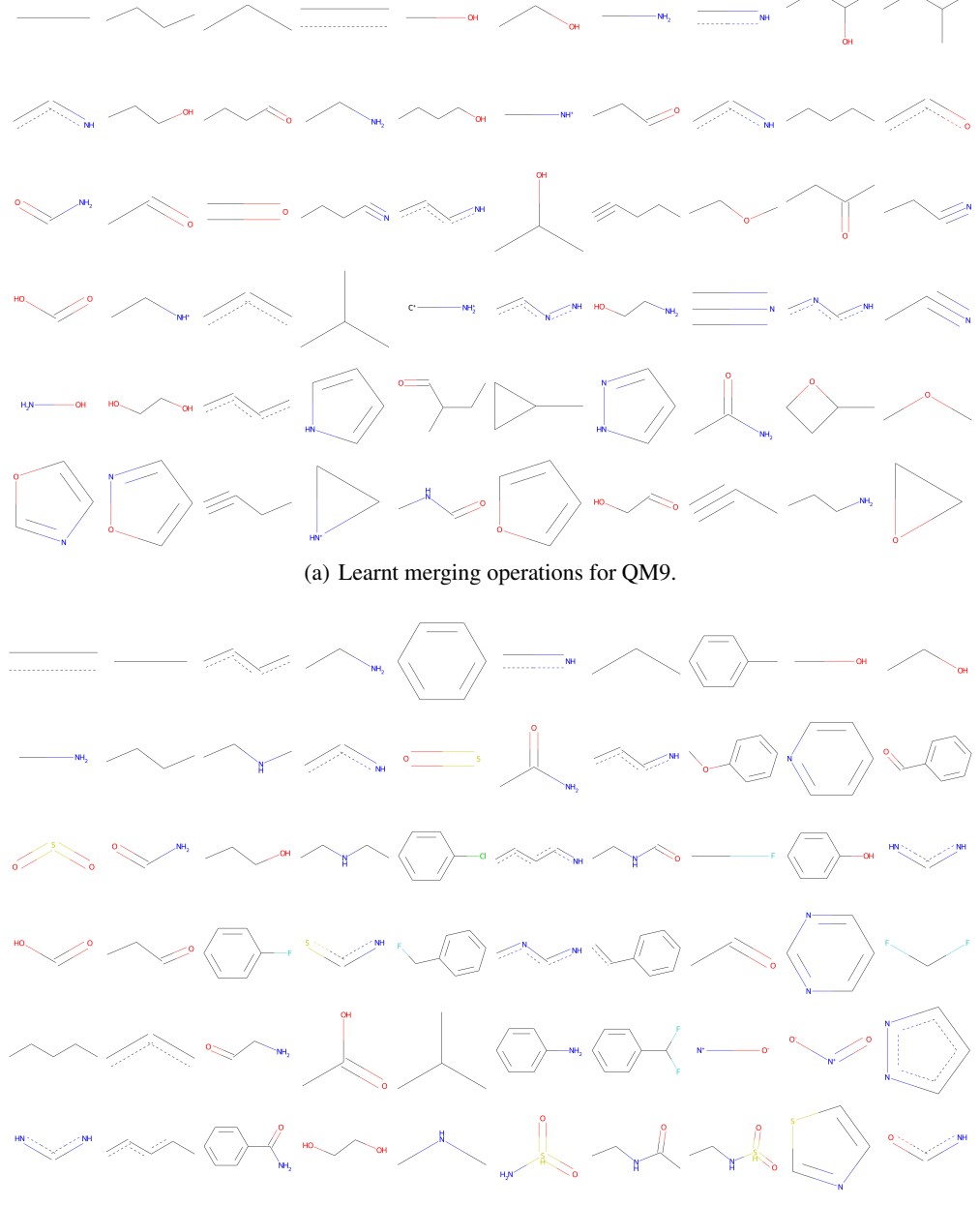

(a) Learnt merging operations for QM9.

(b) Learnt merging operations for GuacaMol.

Figure 12: Merging operations. Note that instead of kekulizing all molecules, we maintain the aromaticity of atoms and bonds. Therefore some of the patters seem half-baked.

## C.2  Mined Motifs

Our algorithm is able to mine common graph motifs with high frequency in a large number of molecules, including some motifs with complex structures and domain specific motifs. We present some mined motifs in Figure C.2.

(a) Mined motifs for QM9.

(b) Mined motifs for GuacaMol.

(c) Mind motifs in the last iteration for Ranolazine MPO benchmark.

Figure 13: Some mined motifs.

## C.3  Motif Representations

Since we apply GNN$_{motif}$ to encode motif representations, instead of viewing motifs as discrete tokens, the learnt motif representations can maintain structural information in the sense that similar motifs have close representations. To show this, we visualize some motif representations in Figure 14.

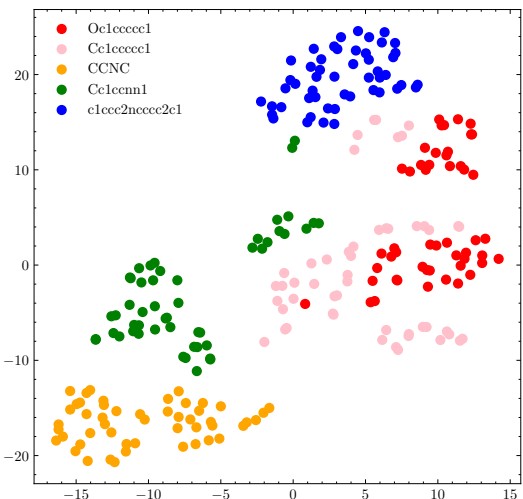

Figure 14: The t-SNE visualization of Motif representations. Each color represents a collection of motifs with a common structure but different connections. Each point represents a motif.

## D   DISCUSSION AND RELATED WORK

**Motif Vocabulary Construction**    Many previous works explored motif construction methods. JT-VAE (Jin et al., 2018) decomposes molecules into rings, chemical bonds, and individual atoms. HierVAE (Jin et al., 2020a) and MoLeR (Maziarz et al., 2021) decompose molecules into ring systems and acyclic linkers or functional groups. MGSSL (Zhang et al., 2021) first uses BRICS (Degen et al., 2008), which is built upon chemical rules, to split molecules into fragments. After that, it manually designs another two rules to further decompose the molecules into rings and chains. Some molecule fragmentation tools such as BRICS and RECAP (Lewell et al., 1998) are also well developed, though outside the ML community. However, as the motif vocabulary obtained by those methods is large and long-tail, the combination of the methods with ML models is nontrivial. The aforementioned methods are mainly based upon pre-defined rules and templates. Guo et al. (2021) proposed DEG, which learns graph grammars to generate molecules and is similar to motif-based methods. However, as DEG applies REINFORCE and MCTS to search grammars, the learned grammars are only for specific metrics, and DEG cannot mine motifs from large datasets. MiCaM mines the most frequent motifs directly from the dataset. The built motif vocabulary is promising to be used in more tasks such as large-scale pre-training, which we leave as future works.

**Goal Directed Generation**    Many frameworks have been developed for goal-directed generation tasks, including iterative methods such as ITA (Yang et al., 2020), genetic methods such as SMILES GA (Yoshikawa et al., 2018) and Graph GA (Jensen, 2019), and latent space optimization methods such as MSO (Winter et al., 2019). RationaleRL (Jin et al., 2020b) proposes to extract rationales from a collection of molecules, and then learns to expand the rationales into full molecules. MolEvol (Chen et al., 2021) then proposes a novel EM-like evolution-by-explanation algorithm bsed on rationales. Such frameworks designed for goal-directed learning tasks can be naturally combined with our proposed generative procedure, which we leave as future works.

