# OpenReview forum: "De Novo Molecular Generation via Connection-aware Motif Mining"
_ICLR.cc/2023/Conference — ICLR 2023 poster_

### Official Review · Reviewer_zHsw · 2022-10-21

**Confidence:** 4
**Correctness:** 3
**Technical Novelty And Significance:** 3
**Empirical Novelty And Significance:** 3
**Recommendation:** 6

**Clarity, Quality, Novelty And Reproducibility:**

# Clarity, Quality
Although the illustration in Figure 2 is intuitive and makes it easy to understand the algorithm, I am confused about the contraction operation (the second iteration in the Merging-operation Learning Phase, i.e., lines 15-17 in Algorithm 1). The algorithm says that for each pair $(\mathcal{F}_i, \mathcal{F}_j) \in \mathcal{E}_M^{(k)}$ (edges in the $k$-th iteration), if $\mathcal{F}_i \oplus \mathcal{F}_j = \mathcal{M}^{(k)}$ then merge them in $\mathcal{G}_M^{(k+1)}$ (the graph at the $(k+1)$-th iteration). What I get confused is that when a pair $(\mathcal{F}_i, \mathcal{F}_j)$ is contracted, then another adjacent pair, say $(\mathcal{F}_j, \mathcal{F}_k) = \mathcal{M}^{(k)}$, may not appear in $\mathcal{G}_M^{(k+1)}$ after the contraction, though it exists still in $\mathcal{E}_M^{(k)}$. This could happen in Figure 2. The original graph has 12 `c:c`s, which can be contracted as in the second left figure in Fig 2 (a), where `c:c` nodes are connected in a triangular shape, but there could be other contraction where we have a square-shape like `(c:c)1:c:(c:c):c:1`. So, I suspect that the merging-operation learning phase is order-dependent, which seems not to be highlighted in the paper. If this is correct, I would appreciate if the authors could discuss about it and how they resolve it; otherwise, I would appreciate if the authors correct my misunderstandings.

# Novelty
As pointed out above, grammar-based methods could be closely related to the proposed method. In order to evaluate the novelty of this work, I would like to hear from the authors the relationship to the grammar-based approaches.

# Reproducibility
This work will be reproducible because the authors provide the source code.

**Strength And Weaknesses:**

# Strengths
- The proposed approach is reasonable and the algorithm is mostly easy to understand (except for the details elaborated in the clarity section of my review).
- In the empirical studies, the authors not only evaluate and compare the performance, but also highlight the properties of the proposed method (Figs. 4 and 5), which are insightful.

# Weaknesses
Relationship to the work by Guo et al. (ICLR 2022), which is cited in the paper, is not clear. As far as I understand, the method proposed by them generates a molecule using a grammar, which can be interpreted as connecting (connection-aware) motifs, and they also optimize the grammar (equivalently, the set of motifs) so as to bias the generative model towards the direction users specify, which implies that the resultant graph grammar is not heuristically designed but is optimal in some sense. Thus, I consider the method by Guo et al. satisfies the two desirable properties suggested in the introduction. I would like to see in-depth discussion on the relationship between their method and the method proposed in this paper, and if they have similar capabilities, they should be compared empirically.

**Summary Of The Paper:**

This paper is concerned about generation of molecules by connecting motifs extracted from the dataset. The main idea of the proposed method is to mine motifs from the dataset. The algorithm, illustrated in Figure 2, first contracts molecular graphs by contracting the most frequent edge, iteratively. Then, motifs are extracted from the contracted graphs as shown in Figure 2 (b), resulting a vocabulary of motifs.

The authors propose a generative model using the vocabulary of motifs. At each iteration of the generation process, a connection site $v_t$ from partial graph $\mathcal{G}_t$ is obtained from the queue, and the other site to be connected is queried by Eq. 1, where the query vector is computed from the partial graph and the input $z$, and the key vector is computed from the other connection site, which may be in the vocabulary or in the partial graph itself. The model is trained in a similar way to VAE, minimizing the loss function in Eq. 3.

The proposed method is evaluated on GuacaMol benchmarks (distributional and goal-directed benchmarks).
For  the distributional learning task, the proposed method achieves relatively higher scores than the baselines. In addition, the authors provide how the scores change as we change the number of merging operations, $K$. The results suggest that as we increase $K$, we get larger motifs, which improves the similarity between the generated molecules and those in the dataset, at the cost of the decreased novelty score, as expected.

For the goal-directed generation result, the proposed model is combined with off-the-shelf optimization modules to generate optimized molecules. The results show that the proposed method achieves better scores than the baseline methods in GuacaMol benchmark. The authors also analyze the generation trajectory to highlight the benefit of utilizing the motif-based approach.

**Summary Of The Review:**

I consider this paper is on borderline, leaning to rejection. While I like the proposed method itself, the relationship between the proposed method and the method by Guo et al. is not clear, which is essential to differentiate the proposed method from the methods in the literature. I would like to re-evaluate the novelty and significance after the authors clarify the relationship.

---

> ### Author Response · Authors · 2022-11-14
> **Response to Reviewer zHsw (1/2)**
>
> We thank the reviewer for the insightful and valuable comments. We respond to your comments as follows and sincerely hope that our rebuttal could properly address your concerns. If so, we would deeply appreciate it if you could raise your score. If not, please let us know your further concerns, and we will continue actively responding to your comments and improving our submission.
>
> **1. Relationship to the work DEG by Guo et al. (ICLR 2022) [1].**
>
> We have added discussions about the grammar-based method DEG proposed by Guo et al. [1] in Appendix D in the revised version. We clarify the differences between our method and DEG from 3 aspects.
>
> - **Tasks.** DEG constructs graph grammars for specific metrics (e.g., diversity and synthesizablity). However, as the grammar construction phase relies on REINFORCE, DEG is hardly directly applicable to distribution learning tasks, where no explicit target metric is provided. MiCaM directly mines motifs from datasets, independent of any explicit metrics, so it is applicable for both distribution learning and goal-directed tasks.
>
> - **Datasets.** DEG aims to deal with situations when the size of the class-specific chemical dataset is limited (e.g., dozens of training samples). However, as DEG relies on a bottom-up search-based method (MCTS) to construct grammars, which is very time costing, we can hardly apply it to large datasets. (As discussed in Section 5.3 in [1], they only deal with a subset of 117 samples when considering a relatively large dataset.) The motif mining algorithm in MiCaM aims to mine the most common motifs from a molecule library, whose size may be very large (e.g., more than 1,200,000 molecules in GuacaMol).
>
> - **Techniques.** The techniques of DEG and MiCaM are quite different. DEG applies a neural network as well as MCTS to construct graph grammars for specific metrics, and then randomly samples grammars to generate molecules without a deep generative model. The constructed grammars can only be used for this specific task. MiCaM directly mines common motifs from the datasets, and then trains a neural network to connect them together. The motif vocabulary built by MiCaM is promising to be combined with deep generative models, and the pre-trained model is useful for more downstream tasks. (e.g., the pre-trained model on the GuacaMol dataset is used for different goal-directed tasks).
>
> Still, we made efforts to have a fair comparison with DEG. We compare MiCaM and DEG using GuacaMol benchmarks.
>
> - **Distribution Learning Results.** We compare the two models on QM9. Following the settings in [1], we sample 117 molecules from QM9 to train DEG (as training more molecules is too time costing) and use diversity and synthesizablity scores as the optimization metrics (we cannot use metrics like KL Divergence or FCD scores, since calculating them requires a large number molecules and is too time costing when training). The results indicate that though DEG generates diverse valid molecules, it does not resemble the distribution of the large dataset very well. We have added the comparison results in Table 1. For your convenience, we quote the results as follows.
>
>   | Model | Validity  | Uniqueness | Novelty   | KL Div    | FCD       |
>   | ----- | --------- | ---------- | --------- | --------- | --------- |
>   | MiCaM | **1.000** | **0.932**  | **0.493** | **0.980** | **0.945** |
>   | DEG   | **1.000** | 0.684      | 0.343     | 0.844     | 0.390     |
>
> - **Goal Directed Results.** We then consider a multi-property objective task, Ranolazine MPO. For DEG, we pick 30 molecules with the highest scores from GuacaMol to learn grammars, and we use the score function as the optimization metric. We then use the learned grammars to generate 10,000 molecules and pick the top 100 ones for benchmarking. Due to the limited rebuttal time and the training speed of DEG, we are only able to run one goal-directed benchmark. We report the results in the following table.
>
>   | Benchmark      | Dataset | DEG   | MiCaM(ours) |
>   | -------------- | ------- | ----- | ----------- |
>   | Ranolazine MPO | 0.792   | 0.817 | **0.932**   |

---

> > ### Author Response · Authors · 2022-11-14
> > **Response to Reviewer zHsw (2/2)**
> >
> > **2. The merging-operation learning phase is order-dependent.**
> >
> > Nice catch! Admittedly the merging-operation learning phase is order-dependent. As we mentioned in Footnote 2, we traverse the edges following the orders given by RDKit. For example, as shown in Figure 2, when the first “c:c” is contracted, another “c:c” disappears, and we will skip the latter. We modify Algorithm 1 to show this detail. For your convenience, we provide the python-style pseudocode of traversing the edges to merge motifs in the end.
> >
> > Being order-dependent does not damage performance. The aforementioned situation only happens when two adjacent pairs are the same, thus merging either of them is reasonable. In fact, an initial ordering of atoms is necessary, which is an intrinsic issue that exists to most graph based generation methods. [2] conducted experiments on the effect of the initial atom orders, and found that: following a random initial order and following the order given by RDKit are similar to the model performance.
> >
> > **Python-style Pseudocode: Traversing the edges to merge motifs.**
> >
> > Input: $ \mathcal{G}_M^{(k)} $, $\mathcal{M}^{(k)}$
> >
> > Output: $\mathcal{G}_M^{(k+1)}$
> >
> > $\mathcal{G}_M^{(k+1)}\leftarrow\mathcal{G}_M^{(k)}$.copy()
> >
> > for $(\mathcal{F}_i,\mathcal{F}_j)$ in $\mathcal{G}_M^{(k)}$.edges:    // following the orders given by RDKit
> >
> > ____if not $\mathcal{G}_M^{(k+1)}$.has_edge($\mathcal{F}_i,\mathcal{F}_j$): continue
> >
> > ____if $\mathcal{F}_i\oplus\mathcal{F}_j$ == $\mathcal{M}^{(k)}$:
> >
> > ________Merge $\mathcal{F}_i$ and $\mathcal{F}_j$ in $\mathcal{G}_M^{(k+1)}$
> >
> >
> > [1] Guo et al., Data-efficient graph grammar learning for molecular generation. (ICLR 2022). https://arxiv.org/abs/2203.08031
> >
> > [2] Maziarz, Krzysztof, et al., Learning to Extend Molecular Scaffolds with Structural Motifs. (ICLR 2021) https://arxiv.org/abs/2103.03864

---

> > ### Comment · Reviewer_zHsw · 2022-11-19
> > **Re:**
> >
> > Thank you very much for clarification and additional results. I understand that the method proposed in this paper has advantages over the work by Guo et al. (ICLR 2022). If MiCaM does not always perform better than DEG, it would be insightful to discuss such cases, which will clarify a limitation of MiCaM.

---

> > > ### Author Response · Authors · 2022-11-21
> > > **Thank you for your kind support.**
> > >
> > > Dear Reviewer zHsw,
> > >
> > > Thanks for your kind support and for helping us improve the paper!
> > >
> > > DEG achieves state-of-the-art results when the size of the class-specific chemical dataset is limited (e.g., dozens of samples). However, the training of MiCaM requires a relatively large molecule library to mine frequent substructures and learn to connect them. We plan to extend our motif mining algorithm to small datasets with only dozens of samples., e.g., by designing a reward function to reinforce the merging operation learning procedure. We leave it as future works.
> > >
> > > As we cannot modify the revision currently, we will add the discussions to clarify the limitation of MiCaM once this paper is accepted.

---

### Official Review · Reviewer_5Fqs · 2022-10-24

**Confidence:** 5
**Correctness:** 2
**Technical Novelty And Significance:** 2
**Empirical Novelty And Significance:** 3
**Recommendation:** 6

**Clarity, Quality, Novelty And Reproducibility:**

This paper is well-written and provides the codebase as supplementary materials for ensuring reproducibility.
As for the novelty, it would be a bit incremental, and would be better to relate this vocabulary construction to the standard fragmentation-based library construction such as BRICS and RECAP in chemoinformatics and that in the previous work [1].

**Strength And Weaknesses:**

[Strength]

- The paper presented an interesting method that generates molecules based on "mined" and "connection-aware" motifs. Due to these considerations, the generator part was effectively designed as a VAE model to just select motifs and determine how they are combined with each other.

- The experiments include two demonstrative tasks of distributional learning results (to produce a set of new molecules that are structurally similar to the given training set) as well as the goal-directed generation tasks (to produce new molecules better than the given training set with respect to the target property) using the GuacaMol benchmarks, which are one of the well-designed benchmarking datasets.

[Weaknesses]

- One significant feature of the proposed method is the way to construct motif vocabulary that is not predefined but is directly mined from the given data, keeping the connection information for further generation uses. But this main claim would sound very incremental when considering a previous work [1]. (Disclaimer: I have no relationship at all with the authors of this paper [1][2][3] below)

The paper [1] uses a traditional chemoinformatics algorithm BRICS [2] with further refinements to construct a motif tree (motifs at nodes and connection information forms a tree hierarchy on motif vocabulary). The authors could check Figure 2 of [1] to describe the motif vocabulary procedure and would find that this traditional procedure can also be pure data-driven ("directly mined from the given dataset") and connection-aware (due to the constructed motif tree), i.e., both "mined" and "connection-aware."

[1] Zhang et al. Motif-based Graph Self-Supervised Learning for Molecular Property Prediction. (NeurIPS'21)
    https://arxiv.org/abs/2110.00987

[2] Degen et al. On the Art of Compiling and Using 'Drug-Like' Chemical Fragment Spaces. (2008)
    http://doi.org/10.1002/cmdc.200800178

BRICS is a well-known way for molecule fragmentation (and further use of it for molecule library construction), and the original paper [2] was highly cited (it was also cited in the GuacaMol paper, for example). Furthermore, the examples of MiCaM motif vocabulary shown in Appendix C.1 would look like those from typical BRICS (or RECAP) fragmentation.

So to claim the novelty of the "mined" and "connection-aware" motifs, it would be highly appreciated to compare the difference and investigate the performance and properties of MiCaM over BRICS motifs (or other motif construction methods). It is already well-established to generate new molecules using BRICS motifs, and it would be informative to check the RDKit document to get a BRICS-based generator.

http://www.rdkit.org/docs/GettingStartedInPython.html#brics-implementation

Getting these building blocks is practically quite important and well-investigated (outside the ML community, though) because they can subsequently be used to construct virtual screening libraries for targeted drug discovery. For example, a tool called eMolFrag [3] was published in a chemoinformatics journal, and in this paper, the authors can check the related literature on motif vocabulary construction and their use for generating new molecules.

[3] Liu et al., Break Down in Order To Build Up: Decomposing Small Molecules for Fragment-Based Drug Design with eMolFrag. (2017)
    https://doi.org/10.1021/acs.jcim.6b00596

- Related to the above point, some ablation study would be needed to confirm that this paper's motif-vocabulary algorithm contributes to the reported good performance by defining the MiCaM generation procedure with existing motif vocabularies other than the proposed one. For example, reporting any concrete examples that can be generated from MiCaM but not likely from MoLeR would be very informative to readers.

- From the reported results of Table 1 (GuacaMol), KL Divergence and FCD were better than MoLeR, but Uniqueness and Novelty were worse than MoLeR. This could also suggest that generated molecules from MiCaM are more like the training set and less valuable in the sense of molecular structure exploration. So it would require a more detailed investigation to systematically understand how this difference is in actual examples.


**Summary Of The Paper:**

This paper develops a novel molecular generation method called MiCaM that simultaneously selects motifs from a motif library and determines how they are connected (or terminate the generation). The key feature of MiCaM is that the motif library (motif vocabulary), the collection of frequent substructural fragments, is built directly from the given set of molecules by iteratively merging subgraphs based on their frequency, also with storing the connection information (this was motivated by byte-pair encoding in MLP). Using this pre-acquired motif library with connection information, MiCaM was trained in a VAE manner to generate novel molecules by selecting motifs and combining them in learned ways. The empirical comparisons to several existing baselines such as JT-VAE, GCPN, and GP-VAE, as well as MoLeR (Maziarz et al., 2021) that generates molecules by extending scaffolds with substructural motifs, demonstrate that it can well resemble the distributions of training sets, keeping high uniqueness and novelty, and it can also perform well on goal-directed benchmarks of the GuacaMol benchmarks.

**Summary Of The Review:**

The paper presented an interesting method that generates molecules by selecting and combining "mined" and "connection-aware" motifs. But possible alternative methods for this vocabulary building, in particular widely ones such as BRICS (also used in the NeurIPS'21 GNN paper [1]), are not evaluated, and the claim of novelty and significance would be unconvincing, at least in its present form.

---

> ### Author Response · Authors · 2022-11-14
> **Response to Reviewer 5Fqs (1/3)**
>
> We thank the reviewer for the insightful and valuable comments. We respond to your comments as follows and sincerely hope that our rebuttal could properly address your concerns. If so, we would deeply appreciate it if you could raise your score. If not, please let us know your further concerns, and we will continue actively responding to your comments and improving our submission.
>
> **Weakness 1.**
> > One significant feature of the proposed method is the way to construct motif vocabulary that is not predefined but is directly mined from the given data, keeping the connection information for further generation uses. But this main claim would sound very incremental when considering a previous work [1].
>
> Thanks for providing the valuable references. Our method is significantly different from them.
>
> - **Differences with [1].**
>
>   - We humbly point out that perhaps you have a different understanding of the word “mined”. **We want to emphasize that MiCaM discovers motifs based on their frequency in the dataset.** In contrast, [1] first uses BRICS, which is built upon chemical rules, to split molecules into fragments. After that, [1] manually designs another two rules to further decompose the molecules into rings and chains. Due to the second step, the motif vocabulary of [1] is essentially similar to that in MoLeR [4] and **only contains rings and chains**. Therefore, the claim in our introduction, that previous algorithms cannot cover complex motif patterns like “FC(F)(F)c1ccccc1”, still holds for [1].
>
>   - Perhaps we understand the word “connection-aware” a little differently. **MiCaM preserves the connection information (i.e., the “\*”s) in the motif vocabulary.** In comparison, [1] does not preserve the “\*”s in the motif vocabulary but applies a neural network to predict the motif tree. Notice that, although there are “\*”s in Figure 2 of [1], the "\*"s are not preserved in the final motif vocabulary. You may want to see the second row of Figure 2 in [1] for examples of their motifs, which are **rings and chains without “\*”s**.
>
> - **Comparison with RECAP and BRICS.**
>
>   - RECAP and BRICS are both well-designed molecule fragmentation strategies defined by chemical rules. **However, directly applying them to ML-based molecular generation tasks is hard because the obtained vocabulary is usually large and long-tail.** As BRICS is an expansion of RECAP, we take BRICS as an example. BRICS breaks molecules into fragments based on 16 chemistry rules. We apply BRICS to QM9, and obtain 77,665 different fragments. Among them, 95.4% only appear one time in the dataset. These limitations are demonstrated in Section 3.1 in [1], and are also the reasons why the authors of [1] have to further decompose the molecules into rings and chains.
>
>   - “The examples of MiCaM motif vocabulary shown in Appendix C.1 would look like those from typical BRICS (or RECAP) fragmentation.” **This is exactly the advantage of our method, which verifies that the motifs that we mine purely from the data tend to be chemically meaningful.** Moreover, due to the aforementioned limitations (i.e., large and long-tail vocabulary), BRICS and RECAP are not directly applicable, while the motifs mined by our method benefit the deep generative model.
>
>   - “So to claim the novelty of the 'mined' and 'connection-aware' motifs, it would be highly appreciated to compare the difference and investigate the performance and properties of MiCaM over BRICS motifs (or other motif construction methods).” Good suggestions! Please refer to our response to Weakness 2 for the comparison.
>
> - **About eMolFrag [3].** Thanks for recommending the paper [3]. In [3], the authors propose eMolFrag to fragmentize molecules, and then apply some rules (eSynth) to combine the fragments together to generate new molecules. **However, the limitations of BRICS, which we discussed in previous response, still exist in eMolFrag and some other fragmentation tools.** Such vocabulary construction methods are mainly originated and oriented for rule-based molecular generation rather than ML models. As the vocabulary obtained by those methods (e.g., BRICS, eMolFrag) is large and long-tail, the combination of the methods with ML models is nontrivial, and we leave it as future works.

---

> > ### Author Response · Authors · 2022-11-14
> > **Response to Reviewer 5Fqs (2/3)**
> >
> > **Weakness 2.**
> >
> > > Related to the above point, some ablation study would be needed to confirm that this paper's motif-vocabulary algorithm contributes to the reported good performance by defining the MiCaM generation procedure with existing motif vocabularies other than the proposed one. For example, reporting any concrete examples that can be generated from MiCaM but not likely from MoLeR would be very informative to readers.
> >
> > - **Ablation studies**
> >
> >   We conduct two more experiments on QM9 to verify the effect of the motif vocabulary. We use the generation procedure of MiCaM, but replace the motif vocabulary with the vocabularies in [1] (BRICS with further decompositions) and MoLeR [4], respectively. For a fair comparison, we preserve the connection information (i.e., the “*”s) in the two new vocabularies. We name the two models as MiCaM-brics and MiCaM-moler, respectively. The results are in **Table 4** in the revised version. For your convenience, we quote **Table 4** as follows.
> >
> >   **Table 4:** Ablation studies on different motif vocabularies on QM9.
> >
> >   | Model       | Validity  | Uniqueness | Novelty   | KL Div    | FCD       |
> >   | ----------- | --------- | ---------- | --------- | --------- | --------- |
> >   | MiCaM-brics | **1.000** | 0.927      | 0.485     | 0.978     | 0.938     |
> >   | MiCaM-moler | **1.000** | 0.926      | 0.468     | 0.973     | 0.934     |
> >   | MiCaM       | **1.000** | **0.932**  | **0.493** | **0.980** | **0.945** |
> >
> > - **Case studies**
> >
> >   We make many efforts to show the differences between the molecules generated by MiCaM and MoLeR.
> >
> >   - We visualize the distributions of molecules generated by MiCaM and MoLeR, respectively. As shown in **Figure 10** in the revised version, MiCaM fits the reference distribution better than MoLeR.
> >   - From the visualization, we find that the outermost contour line of MiCaM covers more area than MoLeR and fits that of the reference data better. This indicates that some reasonable chemical spaces are explored more by MiCaM than MoLeR. We then find that such cases include **molecules with large rings or complex ring systems**. This is because MoLeR only stores simple rings in the motif vocabulary, while MiCaM learns to generate large rings or ring systems from motifs. See **Figure 11** for concrete examples.

---

> > > ### Author Response · Authors · 2022-11-14
> > > **Response to Reviewer 5Fqs (3/3)**
> > >
> > > **Weakness 3.**
> > >
> > > > From the reported results of Table 1 (GuacaMol), KL Divergence and FCD were better than MoLeR, but Uniqueness and Novelty were worse than MoLeR. This could also suggest that generated molecules from MiCaM are more like the training set and less valuable in the sense of molecular structure exploration. So it would require a more detailed investigation to systematically understand how this difference is in actual examples.
> > >
> > > - **There is a trade-off among these metrics.** This is because KL Divergence and FCD negatively correlate with Uniqueness and Novelty. We can achieve this trade-off via some hyperparameters. For example, on QM9, if we conduct 500 merging operations, and use distribution mode with a beam search (sampling from top 5 choices) for sampling, MiCaM will achieve higher uniqueness and novelty, outperforming MoLeR in terms of all metrics. The results are in **Table 3** in the revised version. For your convenience, we quote **Table 3** as follows.
> > >
> > >   **Table 3:** MiCaM can achieve a trade-off among the distribution learning metrics. With 1000 merging operations and greedy mode, MiCaM significantly outperforms MoLeR in terms of KL Divergence and FCD. While with 500 merging operations and distribution mode, MiCaM outperforms MoLeR in terms of all the metrics.
> > >
> > >   | Model             | Validity  | Uniqueness | Novelty   | KL Div    | FCD       |
> > >   | ----------------- | --------- | ---------- | --------- | --------- | --------- |
> > >   | MoLeR             | **1.000** | 0.940      | 0.355     | 0.969     | 0.931     |
> > >   | MiCaM-1000-greedy | **1.000** | 0.932      | 0.493     | **0.980** | **0.945** |
> > >   | MiCaM-500-distr   | **1.000** | **0.941**  | **0.495** | 0.978     | 0.940     |
> > >
> > > - **KL Divergence and FCD are somehow more important.**
> > >   - In drug discovery, our goal is to generate drug-like candidates. Generating novel molecules is relatively easy (e.g., via random sampling), but if the molecules are low-quality, they are usually useless.
> > >   - For ZINC and GuacaMol, Uniqueness and Novelty are high enough (almost 100%), which indicates that most of the generated molecules are unique and novel. An incremental on Uniqueness and Novelty is meaningless. However, the low FCD of previous models indicates that resembling the training set is a more difficult challenge. Compared with MoLeR, MiCaM substantially increases the FCD scores.
> > >
> > > - **MiCaM is good at structure exploration.** MiCaM preserves more structural information in the motif vocabulary than MoLeR, so it is biased to generate molecules with common seen local patterns but novel structures. It does not tend to generate molecules with only minor modifications, e.g., molecules with only a bond or an atom changed. As mentioned in our response to Weakness 2, MiCaM explores wider chemical spaces than MoLeR, covering more molecules with complex structures like large rings and ring systems.
> > >
> > >
> > > **Novelty.**
> > >
> > > > As for the novelty, it would be a bit incremental, and would be better to relate this vocabulary construction to the standard fragmentation-based library construction such as BRICS and RECAP in chemoinformatics and that in the previous work [1].
> > >
> > > Thanks for your suggestions! The novelty of MiCaM is two-fold.
> > >
> > > - **The motif mining algorithm.** We have discussed the relations between our vocabulary construction methods and several previous ones (including MGSSL [1], BRICS[2], eMolFrag[3] and MoLeR [4]. Please refer to the response to Weaknesses 1, 2 and 3.
> > >
> > > - **The connection-aware generation procedure based on the motif vocabulary.** In this procedure, we develop novel techniques to explore better use of the motif vocabulary. We propose to use a GNN$_{\text{motif}}$ to obtain motif representations and select connection sites. As discussed in **Section 2.4**, this helps the model to deal with a large motif vocabulary, generalize to less-seen motifs and preserve the similarity among motifs. Moreover, we design a contrastive learning method for efficient training on a large motif vocabulary.
> > >
> > >
> > > [1] Zhang et al., Motif-based Graph Self-Supervised Learning for Molecular Property Prediction. (NeurIPS'21) https://arxiv.org/abs/2110.00987
> > >
> > > [2] Degen et al., On the Art of Compiling and Using 'Drug-Like' Chemical Fragment Spaces. (2008) http://doi.org/10.1002/cmdc.200800178
> > >
> > > [3] Liu et al., Break Down in Order To Build Up: Decomposing Small Molecules for Fragment-Based Drug Design with eMolFrag. (2017) https://doi.org/10.1021/acs.jcim.6b00596
> > >
> > > [4] Maziarz, Krzysztof, et al., Learning to Extend Molecular Scaffolds with Structural Motifs. (ICLR’21) https://arxiv.org/abs/2103.03864

---

> ### Author Response · Authors · 2022-11-21
> **We are looking forward to your further comments and/or questions.**
>
> Dear Reviewer 5Fqs,
>
> Thanks again for your valuable comments and constructive suggestions, which are of great help to improve the quality of our work. We are looking forward to your further comments and/or questions.
>
> We sincerely hope that our rebuttal has properly addressed your concerns. If so, we would deeply appreciate it if you could raise your score. If not, please let us know your further concerns, and we will continue actively responding to your comments and/or questions.
>
> Best,
>
> Authors

---

> ### Comment · Reviewer_5Fqs · 2022-11-22
> **Thanks for clarification, and sorry for this late post**
>
> Sorry for this late entry. I got the discussion dates mixed up because the initial notification to us was Discussion period: Nov 4 2022 - Dec 12 2022. PC says that authors can still participate in discussion till Dec 12, and so I'll leave my comment here.
>
>
> First of all, thank you for your careful responses. Also, it was really nice to see some updates such as ablation studies.
> I understood the differences with [1] thanks to the clarification! But at the same time, things like "There is a trade-off among these metrics." make me confused about what scores are the reasonable goal of this paper and whether it is not a matter of model tuning. Because the presented method would be a relatively simple approach of combining generated fragments by VAE, seeing strong empirical results is one of advantages of this paper. I would disagree "KL Divergence and FCD are somehow more important." because we also need to check any overfitting to the training dataset or generation redundancy.
>
> - As for BRICS/RECAP fragments, they do not apply to molecular generation because the number of generated fragments are too many? 77,665 different fragments for QM9 (95.4% only appear one time) means that we still have around >3500 fragments if we just select the ones appeared more than twice. These didn't work? How many numbers of fragments the proposed algorithm produced for the same dataset?
>
> The presented fragments are also focused on "frequent" ones, so cutting out infrequent ones might be fair if the claim is "BRICS/RECAP cannot be used because they are too many".
>
> Could we see the result for GuacaMol instead of Table 3?
>
> QM9 originally comes from a subset of the GDB-17 dataset [1] that consists of all possible molecules with nine heavy atoms of C, O, N, and F. So I'm not quite sure this is appropriate for the paper's evaluation purpose. BRICS/RECAP for QM9 might be not informative, but so is the presented one (because they are enumerative in the first place?).
>
> [1] L. Ruddigkeit, R. van Deursen, L. C. Blum, J.-L. Reymond, Enumeration of 166 billion organic small molecules in the chemical universe database GDB-17, J. Chem. Inf. Model. 52, 2864–2875, 2012.

---

> > ### Author Response · Authors · 2022-11-23
> > **Response to additional questions (1/2)**
> >
> > Dear Reviewer 5Fqs,
> >
> > Thanks for your reply and for your valuable comments! We respond to your concerns as follows.
> >
> > > [Q1] But at the same time, things like "There is a trade-off among these metrics." make me confused about what scores are the reasonable goal of this paper and whether it is not a matter of model tuning. Because the presented method would be a relatively simple approach of combining generated fragments by VAE, seeing strong empirical results is one of advantages of this paper. I would disagree "KL Divergence and FCD are somehow more important." because we also need to check any overfitting to the training dataset or generation redundancy.
> >
> > - Please note that **generating novel molecules and resembling the training set are two goals that negatively correlate with each other** [1]. Moreover, some previous works [2, 3] demonstrate that resembling the distribution is an essential task in drug discovery. [4] found that sampling useless molecules by some simple rules will result in state-of-the-art performances in terms of Validity, Novelty, and Uniqueness, but could not obtain competitive scores on FCD. The results in [4] indicate that, though no metrics are perfect, **FCD is a relatively more powerful metric**, at least to some degree.
> > - **MiCaM significantly outperforms baselines in terms of KL Div and FCD on all datasets while keeping competitive in terms of Novelty and Uniqueness.** The Uniqueness of MiCaM on QM9 is slightly lower than MoLeR, but the difference is minor (0.8%). The Uniqueness and Novelty of MiCaM are slightly lower than MoLeR on GuacaMol. However, on GuacaMol, the two metrics are nearly perfect (0.994 and 0.986), implying that almost all molecules are novel and unique, and the model tends not to overfit to the training dataset. This is enough for real applications, and an incremental improvement on them is meaningless.
> > - In real applications in drug discovery, to sample 10,000 unique novel molecules (for GuacaMol), MoLeR needs to decode 10,095 samples, and MiCaM needs to decode 10,164 samples. The difference is only 0.7%. However, the KL Div and FCD scores imply that the molecules generated by MiCaM tend to be more drug-like and useful, which are what drug designers really care about.
> >
> > > [Q2] As for BRICS/RECAP fragments, they do not apply to molecular generation because the number of generated fragments are too many? 77,665 different fragments for QM9 (95.4% only appear one time) means that we still have around >3500 fragments if we just select the ones appeared more than twice. These didn't work?
> >
> > This will not work because of the following reasons.
> >
> > - When training the model, we need to provide ground truth for the model to learn how to reconstruct the molecules in the training set. Therefore, the molecule fragmentation strategy should satisfy a property: an arbitrary molecule can be fragmentized into motifs that belong to the motif vocabulary. In other words, we can reconstruct an arbitrary molecule using the motifs in the vocabulary. However, if we select fragments that appear more than twice to build the motif vocabulary but drop out the fragments that appear only once, **most of the molecules would not be fragmentized into motifs in the vocabulary**.
> >   - For example, it is possible that a molecule is divided into several fragments by BRICS, but one of the fragments only appears once, so it is not in the vocabulary. In this case, the model cannot learn to reconstruct this molecule.
> >   - In addition, BRICS and RECAP are designed from some chemical rules related to chemical reactions. However, **many molecules cannot be finely decomposed by these rules** (i.e., such molecules are viewed as an entire fragment by BRICS instead of being further decomposed), e.g., "N=CNC(=N)C(F)(F)F" and "O=C1CC=C(CO)CN1". When applying BRICS on QM9, we find that 73,053/132,685 (i.e., 55.1%) molecules are such cases. If we drop out fragments that appear once, all these molecules are also dropped out from the training set, as they cannot be generated from the other motifs.
> > - We can ensure the ability to generate molecules in the training dataset by adding the basic atoms into the motif vocabulary, which is also a common practice. However, this means that **the model has to rebuild the structures not in the motif vocabulary atom-by-atom**. For example, for QM9, the model has to learn to generate 55.1% of the molecules in QM9 atom-by-atom. This would fail to take advantages of motif based methods.

---

> > > ### Author Response · Authors · 2022-11-23
> > > **Response to additional questions (2/2)**
> > >
> > > > [Q3] How many numbers of fragments the proposed algorithm produced for the same dataset?
> > >
> > > **MiCaM allows adjusting the number of merging operations, which corresponds to the number of fragments.** For example, if we apply 500 merging operations, each of the merging operations corresponds to a motif pattern, and MiCaM will produce 500 motifs. Moreover, the 500 motifs are frequent in the dataset, so the ML model can learn much about them.
> > >
> > > > [Q4] The presented fragments are also focused on "frequent" ones, so cutting out infrequent ones might be fair if the claim is "BRICS/RECAP cannot be used because they are too many".
> > >
> > > **Though MiCaM focuses on "frequent" fragments, this is essentially different from cutting out infrequent ones.** MiCaM decomposes a molecule into motifs by iteratively merging the most common patterns, and thus each obtained motif is frequent. In contrast, the disadvantages of cutting out infrequent ones have been discussed in our response to [Q2].
> > >
> > > > [Q5] Could we see the result for GuacaMol instead of Table 3?
> > >
> > > Thanks for your suggestion! We report some more results for GuacaMol in the following table. With 500 merging operations and distribution mode, **MiCaM outperforms MoLeR in terms of all the metrics**. For your convenience, we also quote Table 3 (results for QM9) here.
> > >
> > > | Model             | Validity | Uniqueness | Novelty | KL Div | FCD     |
> > > | ----------------- | -------- | ---------- | ------- | ------ | ----- |
> > > | MoLeR             | **1.000** | **1.000**  | 0.991   | 0.964  | 0.625 |
> > > | MiCaM-500-greedy  | **1.000** | 0.994      | 0.986   | **0.989** | 0.731 |
> > > | MiCaM-500-distr   | **1.000** | **1.000** | **0.992** | 0.978 | 0.714 |
> > > | MiCaM-1000-greedy | **1.000** | 0.994      | 0.981   | 0.988  | **0.733** |
> > > | MiCaM-1000-distr  | **1.000** | 0.998     | 0.991  | 0.983 | 0.720 |
> > >
> > > **Table 3:** MiCaM can achieve a trade-off among the distribution learning metrics. With 1000 merging operations and greedy mode, MiCaM significantly outperforms MoLeR in terms of KL Divergence and FCD. While with 500 merging operations and distribution mode, MiCaM outperforms MoLeR in terms of all the metrics.
> > >
> > > | Model             | Validity  | Uniqueness | Novelty   | KL Div    | FCD       |
> > > | ----------------- | --------- | ---------- | --------- | --------- | --------- |
> > > | MoLeR             | **1.000** | 0.940      | 0.355     | 0.969     | 0.931     |
> > > | MiCaM-1000-greedy | **1.000** | 0.932      | 0.493     | **0.980** | **0.945** |
> > > | MiCaM-500-distr   | **1.000** | **0.941**  | **0.495** | 0.978     | 0.940     |
> > >
> > > > [Q6] BRICS/RECAP for QM9 might be not informative, but so is the presented one (because they are enumerative in the first place?).
> > >
> > > We conduct BRICS on QM9 because it is a relatively small dataset. Even on QM9, BRICS produces so many fragments that the model can hardly deal with them. **On GuacaMol, BRICS produces 167,276 fragments**, and 140,230 (83.8%) of them only appear once. Such a large number of fragments is unacceptable for learning a ML model.
> > >
> > > **As we cannot modify the revision at this stage, we will add the above discussions and results for further clarification once this paper is accepted.**
> > >
> > >
> > >
> > > [1] Mahmood O, Mansimov E, Bonneau R, et al. Masked graph modeling for molecule generation[J]. Nature communications, 2021, 12(1): 1-12. https://doi.org/10.1038/s41467-021-23415-2
> > >
> > > [2] Preuer K, Renz P, Unterthiner T, et al. Fréchet ChemNet distance: a metric for generative models for molecules in drug discovery[J]. Journal of chemical information and modeling, 2018, 58(9): 1736-1741. https://pubs.acs.org/doi/abs/10.1021/acs.jcim.8b00234
> > >
> > > [3] Grant L L, Sit C S. De novo molecular drug design benchmarking[J]. RSC Medicinal Chemistry, 2021, 12(8): 1273-1280. https://pubs.rsc.org/en/content/articlehtml/2021/md/d1md00074h
> > >
> > > [4] Renz P, Van Rompaey D, Wegner J K, et al. On failure modes in molecule generation and optimization[J]. Drug Discovery Today: Technologies, 2019, 32: 55-63. https://chemrxiv.org/engage/chemrxiv/article-details/60c74a88567dfe1ae6ec4db1

---

> > ### Comment · Reviewer_5Fqs · 2022-11-24
> > **Thanks, I understood the point and increased my score from 3 to 6**
> >
> > Thank you for the clear answers and additional results. I understand why byte-pair-encoding-like motif building is a perfect fit for the purpose. If we enumerate frequent substructural patterns and ignore infrequent peripherals, we will lose the completeness of the motif vocabulary for rebuilding. So it sounds like that it won't work if we pick any existing motif builder or frequent pattern miner and select some frequent ones.
> >
> > Reviewer zHsw also pointed out this, but this motif-building step reminds us of grammar-based compression. I am a bit familiar with Re-Pair algorithm for string compression rather than byte-pair encoding in NLP (?), and either way, "compression" needs to retain the original information with keeping up the dictionary small, which would be similar to the motivation of this motif building in principle.
> >
> > It is always nice to see any clear and simple method works and the presented empirical results, including informative updates in the discussion phase, strengthen the value of the proposed approach. Thanks for the thorough response to all review comments in the discussion.

---

> > > ### Author Response · Authors · 2022-11-24
> > > **Thank you for your kind support.**
> > >
> > > Dear Reviewer 5Fqs,
> > >
> > > Thanks for your kind support and for helping us improve the paper! We enjoy communicating with you and appreciate your insightful comments.
> > >
> > > Best,
> > >
> > > Authors

---

### Official Review · Reviewer_3hrY · 2022-10-25

**Confidence:** 4
**Correctness:** 3
**Technical Novelty And Significance:** 3
**Empirical Novelty And Significance:** 2
**Recommendation:** 6

**Clarity, Quality, Novelty And Reproducibility:**

Expression of this paper is not good enough, which makes it hard to read.
The method is relatively novel, but the improvement is not significant.


**Details Of Ethics Concerns:**

No ethics concerns.

**Strength And Weaknesses:**

Strength:
  1) Building fragment vocabulary by data mining rather than pre-defining, which can avoid artificial bias.
  2) Selecting new connection sites from both motif vocabulary and the intermediate molecule itself, which allows generating larger rings.
  3) The authors conduct different experiments on various datasets and the problem formulation is clear.

Weaknesses:
  1) The validity rate of the model without chemical validity check was not provided. Some fragments must combine with each other to form chemical groups (e. g. aromatic systems), therefore only depending on data mining to build motif may neglect this entirety. For example, motif ‘Cc1ccnn1’ in Figure 11 is chemically invalid, since it violate the Hückel's rule.
 2) Results in Table 1 indicate that the proposed model did not outperform other baseline models significantly in terms of uniqueness and novelty. The improvements in some metrics for the primary results are very weak, which makes me wonder if the authors' many choices are well justified. The authors could conduct more downstream tasks on the motif-vocabulary to demonstrate the benefits.
 3) How does the model control the size of the molecules generated in the goal directed generation task?
 4) How does the molecule scoring in the Table 2  define?   Can you demonstrate that the improvement comes from the selection of the motif vocabulary?
 5)  Careless writing, e. g. the “ZINC” dataset sometimes is written as “ZINK”.




**Summary Of The Paper:**

This paper tackles the problem of fragment-based molecular generation. The authors propose MiCaM that includes a data-driven algorithm to mine a connection-aware motif vocabulary from a molecule library, as well as a connection-aware generator for de novo molecular generation. For motif vocabulary, the authors evaluated MiCaM on three different datasets and achieved improvement in some metrics. For fragment-based molecular generation,  MiCaM also obtains improvement on goal directed benchmarks.



**Summary Of The Review:**

Expression of this paper is not good enough, which makes it hard to read.
The insight of the proposed model is reasonable but the implementation may neglect chemical rules.

---

> ### Author Response · Authors · 2022-11-14
> **Response to Reviewer 3hrY (1/3)**
>
> We thank the reviewer for the insightful and valuable comments. We respond to your comments as follows and sincerely hope that our rebuttal could properly address your concerns. If so, we would deeply appreciate it if you could raise your score. If not, please let us know your further concerns, and we will continue actively responding to your comments and improving our submission.
>
> **Weakness 1.**
>
> > The validity rate of the model without chemical validity check was not provided. Some fragments must combine with each other to form chemical groups (e. g. aromatic systems), therefore only depending on data mining to build motif may neglect this entirety. For example, motif ‘Cc1ccnn1’ in Figure 11 is chemically invalid, since it violates the Hückel's rule.
>
> - Without a chemical validity check, the validity rates on QM9, ZINC, and GuacaMol are **99.68%, 98.6%, and 98.28%**, respectively. Specifically, as the motifs preserve the connection information, the chemical valence is never violated except for aromatic systems. We provide the results and discussions in **Appendix A.5** in the revised version. Please refer to the third point for further discussions about the invalid cases.
>
> - Although the SMILES “Cc1ccnn1” is chemically invalid, **its connection-aware version is valid**. The legend “Cc1ccnn1” in Figure 11 (**Figure 14** in the revised version) denotes the collection of motifs with a common structure but different connections. Examples of them include “\*-n1:n:c:c:c:1-C” and “\*-n1:n:c:c:c:1-C=\*”, and they do not violate Hückel's rule. We have polished the relevant expression in **Figure 14** in the revised version.
>
> - Still, it is a nice catch to notice that some fragments must combine with each other to form chemical groups (e. g. aromatic systems). For example, it is possible that the model determines to merge two “\*:c:c:c:c:\*”s into “c1ccccccc1”, which is chemically invalid. This only happens when the model generates new aromatic rings. As shown in our previous response, the rates of such invalid cases are low (0.32%, 1.4%, and 1.72% on QM9, ZINC, and GuacaMol, respectively). **Our solution to avoid such invalid cases is:** when the model tries to generate such an invalid aromatic ring, we simply remove the aromaticity of this ring so that the molecule is still valid (e.g., “c1ccccccc1” will be replaced with “C1CCCCCCC1”). There are two alternative solutions:
>
>   - We can kekulize all molecules in the training set so that no aromatic bond will appear. In this case, the validity is absolutely guaranteed. Though this is a useful trick and has been used in many previous works [1, 2], the mined motifs will lose information about the aromaticity, so we did not adopt this trick currently.
>   - We can further design rules to restrict decoding vocabularies and set the probabilities of invalid choices as zero.
>
>   Due to the limited rebuttal time, we cannot run all experiments. However, the effectiveness of our method has been demonstrated, and we will keep working on those potential alternative solutions.

---

> > ### Author Response · Authors · 2022-11-14
> > **Response to Reviewer 3hrY (2/3)**
> >
> > **Weakness 2.**
> >
> > > Results in Table 1 indicate that the proposed model did not outperform other baseline models significantly in terms of uniqueness and novelty. The improvements in some metrics for the primary results are very weak, which makes me wonder if the authors' many choices are well justified. The authors could conduct more downstream tasks on the motif-vocabulary to demonstrate the benefits.
> >
> > - **Uniqueness and Novelty.**
> >
> >   - **MiCaM significantly outperforms baselines in terms of KL Div and FCD on all datasets while keeping competitive in terms of Uniqueness and Novelty.** The Uniqueness of MiCaM on QM9 is slightly lower than MoLeR, but the difference is minor (0.8%). The Uniqueness and Novelty of MiCaM are slightly lower than MoLeR on GuacaMol. However, on GuacaMol, the two metrics are nearly perfect (0.994 and 0.986), implying that almost all molecules are novel and unique, and the model tends not to overfit to the training dataset. This is enough for real applications, and an incremental improvement on them is meaningless.
> >
> >   - In real applications in drug discovery, to sample 10,000 unique novel molecules (for GuacaMol), MoLeR needs to decode 10,095 samples, and MiCaM needs to decode 10,164 samples. The difference is only 0.7%. **However, the KL Div and FCD scores imply that the molecules generated by MiCaM tend to be more drug-like and useful, which are what drug designers really care about.**
> >
> >   - **Please note that generating novel molecules and resembling the training dataset are two goals that negatively correlate with each other.** [1] MiCaM can significantly outperform baselines in terms of KL Div and FCD on all datasets while keeping competitive in terms of Novelty and Uniqueness. Moreover, **MiCaM can also outperform the baselines in terms of all metrics.** For example, on QM9, if we conduct 500 operations, and use distribution mode (sampling from top 5 choices), MiCaM achieves higher uniqueness and novelty that outperform MoLeR. The results are in **Table 3** in the revised version. For your convenience, we quote **Table 3** as follows.
> >
> >     **Table 3:** MiCaM can achieve a trade-off among the distribution learning metrics. With 1000 merging operations and greedy mode, MiCaM significantly outperforms MoLeR in terms of KL Divergence and FCD. While with 500 merging operations and distribution mode, MiCaM outperforms MoLeR in terms of all the metrics.
> >
> >     | Model             | Validity  | Uniqueness | Novelty   | KL Div    | FCD       |
> >     | ----------------- | --------- | ---------- | --------- | --------- | --------- |
> >     | MoLeR             | **1.000** | 0.940      | 0.355     | 0.969     | 0.931     |
> >     | MiCaM-1000-greedy | **1.000** | 0.932      | 0.493     | **0.980** | **0.945** |
> >     | MiCaM-500-distr   | **1.000** | **0.941**  | **0.495** | 0.978     | 0.940     |
> >
> >   - **MiCaM is good at exploring novel structures.**
> >
> >     - MiCaM preserves more structural information in the motif vocabulary than MoLeR, so it is biased to generate molecules with common seen local patterns but novel structures. It does not tend to generate molecules with only minor modifications, e.g., molecules with only a bond or an atom changed.
> >     - To see this, we visualize the probability distributions of GuacaMol, the molecules generated by MiCaM and MoLeR, respectively in Figure 10 in the revised version. We find that the outermost contour line of MiCaM covers more area than MoLeR and fits that of the reference data better. This indicates that some reasonable chemical spaces are explored more by MiCaM than MoLeR. We then find that such cases include molecules with large rings or complex ring systems. See Figure 11 for some concrete examples.
> >
> >
> >
> > - **The benefits of the motif vocabulary.**
> >
> >   We conduct two more experiments on QM9 to verify the effect of the motif vocabulary. Specifically, we use the generation procedure of MiCaM, but replace the motif vocabulary with the vocabularies in MoLeR [2] and MGSSL [3] (which uses an algorithm called BRICS with further decomposition), respectively. For a fair comparison, we preserve the connection information (i.e., the ``*"s) in the two new vocabularies. We name the two models MiCaM-moler and MiCaM-brics, respectively. The results are in Table 4. For your convenience, we quote Table 4 as follows.
> >
> >   **Table 4:** Ablation studies on different motif vocabularies on QM9.
> >
> >   | Model       | Validity  | Uniqueness | Novelty   | KL Div    | FCD       |
> >   | ----------- | --------- | ---------- | --------- | --------- | --------- |
> >   | MiCaM-moler | **1.000** | 0.926      | 0.468     | 0.973     | 0.934     |
> >   | MiCaM-brics | **1.000** | 0.927      | 0.485     | 0.978     | 0.938     |
> >   | MiCaM       | **1.000** | **0.932**  | **0.493** | **0.980** | **0.945** |

---

> > > ### Author Response · Authors · 2022-11-14
> > > **Response to Reviewer 3hrY (3/3)**
> > >
> > > **Weakness 3.**
> > >
> > > > How does the model control the size of the molecules generated in the goal directed generation task?
> > >
> > > We are not sure whether you are talking about the size of one molecule, or the number of molecules we generate in the iterative optimization. Therefore, we respond to both of them.
> > >
> > > - **The size of one molecule.** In goal-directed generation tasks, the scoring functions are well designed by GuacaMol [4], so we do not need to explicitly control the sizes of molecules. Too large molecules can hardly achieve high scores.
> > >
> > >   Moreover, following the common practice in previous works [2, 5], we set a maximum generation steps and force the model to stop when achieving this maximum. Then the non-terminal bonds will be connected with a carbon atom (i.e., “C”), and non-ring aromatic atoms will be removed the aromaticity. We set this number to 20, and find that more than 99.82% of the molecules stop before 20 steps, which indicates most of generated molecules have reasonable sizes.
> > >
> > > - **The number of molecules.** The number of molecules generated in each iteration, the number of molecules stored in the training buffer, and the number of iterations as fixed hyperparameters. Following [6, 7], in this work, we empirically set the three numbers as 80,000, 10,000 and 7, respectively.
> > >
> > > **Weakness 4.**
> > >
> > > > How does the molecule scoring in the Table 2 define? Can you demonstrate that the improvement comes from the selection of the motif vocabulary?
> > >
> > > - **Scoring function definition.**
> > >   - The scoring functions are defined in GuacaMol benchmarks [4]. Briefly, the scoring function of a molecule is defined as the combination of one or several functions that represent different molecular features. The final benchmark score is calculated as a weighted average of the molecule scores.
> > >   - We take the Ranolazine MPO benchmark as an example. The scoring function of a molecule is defined as the geometric mean of four functions: 1. the similarity between the molecule and Ranolazine, 2. logP, 3. TPSA and 4. The number of fluorine atoms. The four functions are computed by RDKit and then modified so that they fall in the interval [0,1]. The final benchmark score is defined as the weighted average of top-1, top-10, and top-100 scores. For more details, we recommend you to refer to Section 7.3 of GuacaMol [4].
> > >
> > > - **The improvement comes from the selection of the motif vocabulary.**
> > >   - We conduct two case studies in **Figure 5** and **Figure 8**, respectively to demonstrate the effectiveness of the motif vocabulary. In Figure 5, we show that the four scores in Ranolazine MPO task increase as some key motifs are added to the molecule, which implies that the picked motifs are relevant to the target properties. In Figure 8, we show that as the number of iterations increases, the motifs learnt by MiCaM tend to be more specific and more adaptive to the target, leading to the model to generate molecules with higher scores while costing fewer generation steps.
> > >   - We conduct two more experiments to verify the effect of the motif vocabulary. Please refer to our response to Weakness 2.
> > >
> > > **Weakness 5.**
> > >
> > > > Careless writing, e. g. the “ZINC” dataset sometimes is written as “ZINK”.
> > >
> > > - Thanks for your kind suggestions. We have corrected the typos and the presentation in the revised paper accordingly.
> > >
> > > [1] Mahmood O, Mansimov E, Bonneau R, et al. Masked graph modeling for molecule generation[J]. Nature communications, 2021, 12(1): 1-12. https://doi.org/10.1038/s41467-021-23415-2
> > >
> > > [2] Maziarz K, Jackson-Flux H R, Cameron P, et al. Learning to Extend Molecular Scaffolds with Structural Motifs[C]//International Conference on Learning Representations. 2021. https://arxiv.org/abs/2103.03864
> > >
> > > [3] Zhang Z, Liu Q, Wang H, et al. Motif-based graph self-supervised learning for molecular property prediction[J]. Advances in Neural Information Processing Systems, 2021, 34: 15870-15882. https://arxiv.org/abs/2110.00987
> > >
> > > [4] Nathan Brown, Marco Fiscato, Marwin HS Segler, and Alain C Vaucher. Guacamol: benchmarking models for de novo molecular design. Journal of chemical information and modeling, 59(3): 1096–1108, 2019. https://pubs.acs.org/doi/10.1021/acs.jcim.8b00839
> > >
> > > [5] Jin W, Barzilay R, Jaakkola T. Hierarchical generation of molecular graphs using structural motifs[C]//International conference on machine learning. PMLR, 2020: 4839-4848. https://proceedings.mlr.press/v119/jin20a.html
> > >
> > > [6] Robin Winter, Floriane Montanari, Andreas Steffen, Hans Briem, Frank Noe, and Djork-Arn ´ e Clev- ´ ert. Efficient multi-objective molecular optimization in a continuous latent space. Chemical science, 10(34):8016–8024, 2019. https://pubs.rsc.org/en/content/articlehtml/2019/sc/c9sc01928f
> > >
> > > [7] Kevin Yang, Wengong Jin, Kyle Swanson, Regina Barzilay, and Tommi Jaakkola. Improving molecular design by stochastic iterative target augmentation. In International Conference on Machine Learning, pp. 10716–10726. PMLR, 2020. http://proceedings.mlr.press/v119/yang20e.html

---

> ### Author Response · Authors · 2022-11-21
> **We are looking forward to your further comments and/or questions.**
>
> Dear Reviewer 3hrY,
>
> Thanks again for your valuable comments and constructive suggestions, which are of great help to improve the quality of our work. We are looking forward to your further comments and/or questions.
>
> We sincerely hope that our rebuttal has properly addressed your concerns. If so, we would deeply appreciate it if you could raise your score. If not, please let us know your further concerns, and we will continue actively responding to your comments and/or questions.
>
> Best,
>
> Authors

---

> > ### Comment · Reviewer_3hrY · 2022-11-25
> > **Thanks for the response**
> >
> > Thank you for your responses and revisions. Most weakness points have been addressed and I have raised the score.
> > Structures that connects to “n1nccc1” with single bond(s) still violate the Hückel's rule as shown in Figure 14. According to the syntax of SMILES, lower-case letters indicate atoms in aromatic systems. However, “n1nccc1” does not form an aromatic system since it contains only 5 π delocalized electrons rather than 4k+2 (k=0, 1, ...). Maybe you mistake the SMILES of the aromatic molecule pyrazole, which can be represented by “n1[nH]ccc1”.

---

> > > ### Author Response · Authors · 2022-11-28
> > > **Thank you for your kind support.**
> > >
> > > Dear Reviewer 3hrY,
> > >
> > > Thanks for your kind support and the helpful discussions about Hückel's rule.
> > >
> > > - In the next version, we will further clarify that "each legend in Figure 14 corresponds to a collection of motifs with a common substructure but different connections". Moreover, to avoid being confusing, we will also replace the current example with a more representative one, pyrazole (“c1cn[nH]c1”).
> > >
> > > - Actually, "\*-n1cccn1" is valid once "\*" is properly chosen. Examples included "Cln1cccn1", (1-Chloro-1H-pyrazole, https://pubchem.ncbi.nlm.nih.gov/compound/19734875, where "\*" is the “Cl”), "Cn1cccn1" (1-Methylpyrazole, https://pubchem.ncbi.nlm.nih.gov/compound/70255, where "\*" is the "C"), and "OC(=O)c1ccnn1-c1cccnc1" (2-pyridin-3-ylpyrazole-3-carboxylic acid, https://pubchem.ncbi.nlm.nih.gov/compound/139271208, where "\*" is the "c1cccnc1"). All the above examples not only exist in PubChem, but also can be parsed into molecules by RDKit.
> > >
> > > - One special case is that when a pyrazolyl nitrogen (i.e., "n") is connected with a hydrogen (i.e., "H"), the nitrogen is written as "[nH]" to distinguish this kind of nitrogen from a pyridyl-N (e.g., the pyrazole “c1cn[nH]c1”). More details are available in Section 3.4.3 of https://www.daylight.com/dayhtml/doc/theory/theory.smiles.html. It is about the expression of aromatic nitrogen compounds, and thank you for pointing out this confusion. We will make revisions as we stated in the first item.
> > >
> > > Best,
> > >
> > > Authors

---

### Official Review · Reviewer_Gyc8 · 2022-10-28

**Confidence:** 3
**Correctness:** 4
**Technical Novelty And Significance:** 4
**Empirical Novelty And Significance:** 4
**Recommendation:** 8

**Clarity, Quality, Novelty And Reproducibility:**

This paper is written clearly. The paper provided implementation details and code for reproducibility. Refer to the previous question for Quality and novelty.

**Strength And Weaknesses:**

Strength:
- The paper presented a Byte-Pair-Encoding-like motif mining algorithm for molecules and demonstrated its usage together with a VAE-based generative model using the mined motifs as building blocks. To my knowledge, such a subgraph mining method is novel and could be used as a generic graph generative model.
- The pointer mechanism in Equation (1) allows the model to select from a set of candidates with varying sizes. Compared with using a trainable embedding vector for each motif, using a GNN to embed the motifs allows the model to generalize to less seen motifs. Similarly, such a method allows the sharing of information between different connection sites.
- Negative sampling (or contrastive learning) is employed to reduce computational costs during training.
- The distributional learning experiment demonstrated the competitive performance of the proposed method.

Weaknesses:
- Lack of baseline comparisons in the goal-directed generation experiments.
- Several motif-based goal-directed molecule generation methods are not included and discussed in the paper. Jin, Wengong, Regina Barzilay, and Tommi Jaakkola. "Multi-objective molecule generation using interpretable substructures." International conference on machine learning. PMLR, 2020. Chen, Binghong, et al. "Molecule optimization by explainable evolution." International Conference on Learning Representation (ICLR). 2021.


**Summary Of The Paper:**

This paper presented a novel fragment (motif) based generative model for molecules. The motif vocabulary is mined from the molecule dataset by iterative merging small motifs into larger ones and keeping the most frequent ones from each iteration. Then a generative model is learned to construct the motifs into molecules. In each generation step, the model embeds the motif candidates, the connection site, and the partial molecule into the embedding space and uses a pointer mechanism to select the next motif to connect to the site. The proposed method achieved competitive distributional learning performance on several molecule datasets. It can also be used in goal-directed generation tasks.

**Summary Of The Review:**

This paper presented a novel fragment-based generative model for molecules. Empirically this method achieved competitive distributional learning performance. Overall the contribution of this paper is solid.

---

> ### Author Response · Authors · 2022-11-14
> **Response to Reviewer Gyc8**
>
> We thank the reviewer for the positive and insightful comments. We respond to your comments as follows and sincerely hope that our rebuttal could properly address your concerns. If so, we would deeply appreciate it if you could raise your score or confidence. If not, please let us know your further concerns, and we will continue actively responding to your comments and improving our submission.
>
> **Weakness 1. Lack of baseline comparisons in the goal-directed generation experiments.**
>
> Thanks for your suggestion! We have added some discussions about the baseline comparisons, as well as two more baselines ([1] and [2]) in Section 3.2 and Appendix D in the revised version. Please refer to our response to Weakness 2 for more details.
>
>
>
> **Weakness 2. Several motif-based goal-directed molecule generation methods are not included and discussed in the paper.**
>
> Thanks for your suggestions! We have added the two baselines RationaleRL[1] and MolEvol[2] on GuacaMol benchmarks.
>
> [1] first extracts rationales (i.e., substructures that are likely responsible for a specific property) from a collection of molecules, and then learns to expand the rationales into full molecules. [2] proposes a novel EM-like evolution-by-explanation algorithm based on the rationales. These two frameworks are designed for goal-directed learning tasks and could be combined with our proposed generative procedure, which we leave as future works. We run the released code of these two baselines. With their default hyperparameters, no rationale is discovered on some benchmarks, and thus the methods will not work. Therefore, we further tune the hyperparameters so that the models can find rationales from the training sets.
>
> We have added the results in Table 2 in the revised version. For your convenience, we report the results as follows.
>
> | Benchmark               | Dataset | RationaleRL | MolEvol   | MiCaM(ours) |
> | ----------------------- | ------- | ----------- | --------- | ----------- |
> | Celecoxib Rediscovery   | 0.505   | **1.000**   | **1.000** | **1.000**   |
> | Aripiprazole Similarity | 0.595   | **1.000**   | **1.000** | **1.000**   |
> | C11H24 Isomers          | 0.684   | 0.983       | 0.988     | **0.999**   |
> | Ranolazine MPO          | 0.792   | 0.916       | 0.921     | **0.932**   |
> | Sitagliptin MPO         | 0.509   | 0.865       | 0.873     | **0.914**   |
>
>
>
> [1]  Jin, Wengong, Regina Barzilay, and Tommi Jaakkola. "Multi-objective molecule generation using interpretable substructures." International conference on machine learning. PMLR, 2020.
>
> [2] Chen, Binghong, et al. "Molecule optimization by explainable evolution." International Conference on Learning Representation (ICLR). 2021.

---

> > ### Comment · Reviewer_Gyc8 · 2022-11-23
> > **Thanks for the response**
> >
> > Thank you for answering my questions and adding more baselines to the experiment! I would maintain my current score and recommend for acceptance.

---

> > > ### Author Response · Authors · 2022-11-24
> > > **Thank you for your kind support.**
> > >
> > > Dear Reviewer Gyc8,
> > >
> > > Thanks for your kind support and for helping us improve the paper! We appreciate your valuable suggestions.
> > >
> > > Best,
> > >
> > > Authors

---

> ### Author Response · Authors · 2022-11-21
> **We are looking forward to your further comments and/or questions.**
>
> Dear Reviewer Gyc8,
>
> Thanks again for your valuable comments and constructive suggestions, which are of great help to improve the quality of our work. We are looking forward to your further comments and/or questions.
>
> We sincerely hope that our rebuttal has properly addressed your concerns. If so, we would deeply appreciate it if you could raise your score or confidence. If not, please let us know your further concerns, and we will continue actively responding to your comments and/or questions.
>
> Best,
>
> Authors

---

### Author Response · Authors · 2022-11-18
**We are looking forward to your further comments and/or questions.**

Dear Reviewers,

Thanks again for your valuable comments and constructive suggestions, which are of great help to improve the quality of our work. As the rebuttal phase is approaching (due on November 18), we are looking forward to your further comments and/or questions.

We sincerely hope that our rebuttal has properly addressed your concerns. If so, we would deeply appreciate it if you could raise your scores. If not, please let us know your further concerns, and we will continue actively responding to your comments and improving our submission.

Best,

Authors

---

### Decision · Program_Chairs · 2023-01-20

**Decision:**

Accept: poster

**Justification For Why Not Higher Score:**

Although the proposed method is well-designed, as I have stated in the meta-review, the novelty is not very high. This means that the core idea is not so inspiring for further development of studies in this field. Based on my opinions and reviewers' scores, Accept with poster is appropriate to this paper.


**Justification For Why Not Lower Score:**

This paper is well written and the proposed method is well-designed and its performance is superior to other state-of-the-art methods under the standard benchmarks (GuacaMol). There are no crucial concerns and all the reviewers agree with accepting the paper. Therefore I accept the paper.

**Metareview: Summary, Strengths And Weaknesses:**

This paper proposes a novel motif-based molecular generation algorithm. The proposal first mines motifs of molecules from a database by iteratively merging smaller motifs based on their frequency. It then trains a VAE-based generative model on the mined motive vocabulary. The performance of the proposed method, called MiCaM, is empirically evaluated on real-world datasets.

### Strength

- The idea of directly building the motif vocabulary from a molecular database by mining frequently appearing motifs and effectively generating molecules using such motifs is interesting. In addition to this interesting main idea, the proposed algorithm is carefully designed with a number of well-prepared techniques.

- Empirical evaluation is thorough and the performance of the proposed algorithm is convincing. Although reviewers pointed out several issues, the authors performed additional experiments to address them in their responses and the revised paper.

- Presentation is good. This paper is well written and easy to follow.

### Weakness

- The novelty is not very high, as the proposal is a combination of existing techniques.

- In the goal directed benchmarks, currently five molecules are tested in the paper, while there are many more goal directed benchmark molecules in GuacaMol. Thus evaluation on other molecules would be valuable.

Overall I think this paper is very well written and most of the concerns raised by reviewers have been successfully addressed in the authors' responses and the revised paper. Therefore it deserves to be accepted in my opinion.

By the way, the paper format in P.3 looks strange (main text is displayed at half size) if the file is opened by Preview on MacOS (there is no problem if opened by Adobe Acrobat). It would be better to check this issue as many people will read your paper after it gets accepted and some of them will use Preview.


**Note From Pc:**

if the above contains the word "oral" or "spotlight" please see: "oral" presentation means -> notable-top-5% and "spotlight" means -> notable-top-25%. As stated in our emails, we are disassociating presentation type from AC recommendations